# BackBench: Are Vision Language Models Resilient to Object-to-Background Context?

## Abstract

In this paper, we evaluate the resilience of modern vision and multimodal foundational models against object-to-background context variations. The majority of robustness evaluation methods have introduced synthetic datasets to induce changes to object characteristics (viewpoints, scale, color) or utilized image transformation techniques (adversarial changes, common corruptions) on real images to simulate shifts in distributions. Our approach, on the other hand, can change the background of real images using text prompts thus allowing diverse changes to the background. We achieve this while preserving the original appearance and semantics of the object of interest. This allows us to quantify the role of background context in understanding the robustness and generalization of deep neural networks. To achieve this goal, we harness the generative capabilities of text-to-image, image-to-text, and image-to-segment models to automatically generate a broad spectrum of object-to-background changes. By using textual guidance for control, we produce various versions of standard vision datasets (ImageNet, COCO), incorporating either diverse and realistic backgrounds into the images or introducing variations in the color and texture of the background. Additionally, we craft adversarial backgrounds by optimizing the latent variables and text embeddings within text-to-image models. We conduct thorough experimentation and provide an in-depth analysis of the robustness of vision and language models against object-to-background context variations across different tasks. Our code and evaluation benchmark along with the datasets will be publicly released.

## 1 Introduction

Deep learning-based vision models have achieved significant improvement in diverse vision tasks. However, the performance on static held-out datasets does not capture the diversity of different object background compositions present in the real world. In order for these models to be deployed in security-critical applications, analyzing the robustness of these models under diverse changes in the distribution of the data is crucial. Previous works have shown that vision models are vulnerable to a variety of image alterations, including common corruptions (e.g., snow, fog, blur) (Hendrycks & Dietterich, 2019; Moayeri et al., 2022), domain shifts (e.g., paintings, sketches, cartoons)(He et al., 2016; Hendrycks et al., 2021a), and changes in viewpoint (e.g., pose, shape, orientation) (Chang et al., 2015; Idrissi et al., 2022; Bordes et al., 2023). Additionally, carefully designed perturbations can be added to images to create adversarial examples that are imperceptible to humans but can fool the decision-making of vision models (Szegedy et al., 2013; Goodfellow et al., 2014).

Several approaches have been proposed to improve the out-of-distribution robustness of vision models. Madry et al. (2017) propose to train the models on adversarial examples in order to achieve adversarial robustness. (Zhang et al., 2017; Cubuk et al., 2018; Yun et al., 2019; Hendrycks et al., 2021a) propose augmentation policies to improve the non-adversarial robustness of models. More recently, the computer vision field has seen the emergence of large-scale pretraining of both vision (Oquab et al., 2023; Kirillov et al., 2023) and vision-language models (Radford et al., 2021; Sun et al., 2023; Li et al., 2023a). These models, train on extensive datasets and multiple modalities, have demonstrated promising performance on non-adversarial distribution shifts due to the rich representation space learned by training at scale. Consequently, several works (Zhou et al., 2022; Khattak et al., 2023) have adapted these models for downstream tasks by utilizing learnable prompts to preserve the rich feature space learned during pre-training.

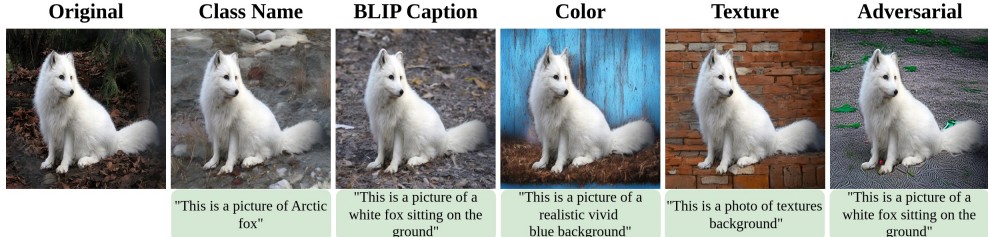

Figure 1: The image-to-background variations by our approach. Each column in the figure represents a specific background generated through the corresponding prompt mentioned below of each image.

To evaluate the vision models on different distribution shifts, numerous datasets, comprising either synthetic or altered real images have been proposed. While synthetic datasets (Chang et al., 2015; Johnson et al., 2017; Gondal et al., 2019) offer more control on variation of scene in the image (background, shape, size, viewpoint), they lack in realism, with most datasets capturing only simple shape objects in a controlled environment. On the other hand, many studies (Hendrycks & Dietterich, 2019; Moayeri et al., 2022) opt for applying coarse-grained image manipulations on the available ImageNet dataset (Deng et al., 2009). This provides more realism but lacks in terms of the diversity in the scene. A recent work (Bordes et al., 2023), tries to bridge this gap by manually creating a synthetic dataset using a powerful game engine to introduce more realism in the images.

In this work, our goal is to introduce diverse background shifts in real images to understand the robustness of vision models from the perspective of object-to-background context. We accomplish this by utilizing the generative capabilities of a text-to-image diffusion model for image editing (Ho et al., 2020; Rombach et al., 2022). Our approach preserves the semantics of the original object (Figure 1) by conditioning the diffusion process on object boundaries and textual descriptions generated by foundational image-to-segment (Kirillov et al., 2023) and image-to-text (Li et al., 2023a) models. We guide the diffusion process to add variations in the background by adding the desired change in the textual description of the image. Additionally, our approach can generate adversarial backgrounds in the feature space of diffusion models by optimizing both its latent variable and text embedding. Thus, our approach allows to generate datasets covering diverse background changes on a selected subset of ImageNet(Deng et al., 2009) and COCO validation set (Lin et al., 2015), enabling comprehensive evaluation of modern unimodal and multimodal models across different tasks. Our contributions are as follows:

- **Framework for Object-to-Backgroud Manipulations.** We propose an automated framework to add diverse background changes to real images, allowing us to benchmark the resilience of modern vision models against object-to-background context.
- **BackBench Dataset.** By carefully filtering the images from ImageNet and COCO validation set, we generate diverse background variations of these datasets by utilizing textual guidance for depicting the desired change.
- **New Benchmark for Resilience Evaluation**. We provide thorough analysis and insights on different unimodal and multimodal models for the task of classification, segmentation, detection, and captioning. Our analysis indicates the recent foundational models are vulnerable to non-adversarial background changes even when the original object is fully preserved in the modified image.

## 2 RELATED WORK

**Corruptions.** Zhu et al. (2016) curate distinct datasets by separating foreground and background elements using ImageNet-1k bounding boxes. They found that models could achieve high object classification performance even when the actual object was absent. Similarly, Rosenfeld et al. (2018) demonstrate that subtle changes in object positioning could significantly impact the detector's predictions, highlighting the sensitivity of these models to spatial configurations. A related approach by Shetty et al. (2019) focuses on co-occurring objects within an image and investigates if removing one object affected the response of the target model toward another. Xiao et al. (2020) analyze

the models' reliance on background signals for decision-making by training on various synthetic datasets. Hendrycks & Dietterich (2019) benchmark the robustness of classifiers against common corruptions and perturbations like fog, blur, and contrast variations. In subsequent work, Hendrycks et al. (2021b) create the ImageNet-A dataset, filtering natural adversarial examples from a subset of ImageNet to limit spurious background cues. Also, Hendrycks et al. (2021a) introduce the ImageNet-R dataset, which comprises various renditions of object classes under diverse visual representations such as paintings, cartoons, embroidery, sculptures, and origami. Similarly, Moayeri et al. (2022) introduce the RIVAL10 dataset to study Gaussian noise corruptions in the foreground, background, and object attributes.

**Viewpoint Changes.** (Chang et al., 2015; Gondal et al., 2019; Alcorn et al., 2019) introduce a large-scale 3D shape datasets to study object scale and viewpoints variations. In a similar vein, Johnson et al. (2017) introduce a synthetic dataset of rendered objects to aid in diagnostic evaluations of visual question-answering models. Later works have made strides in addressing the realism gap, as seen in Idrissi et al. (2022) and Barbu et al. (2019). Barbu et al. (2019) utilize crowdsourcing to control rotation, viewpoints, and backgrounds of household objects, while Idrissi et al. (2022) provide more fine-grained annotations for variations on the ImageNet validation set. In a recent development, Bordes et al. (2023) release a dataset rendered using Unreal Engine under diverse conditions, including varying sizes, backgrounds, camera orientations, and light intensities.

**Adversarial and Counterfactual Manipulations.** Researchers have uncovered that subtle, carefully designed alterations to an image, imperceptible to human observers, have the ability to deceive deep learning models (Szegedy et al., 2013; Goodfellow et al., 2014; Kurakin et al., 2016). These perturbations, constructed using gradient-based methods, serve as a worst-case analysis in probing the model's robustness within specified distance norm metrics ($l_2$ or $l_\infty$). Another strategy entails applying unbounded perturbations to specific image patches, thereby conserving object semantics while inducing model confusion (Sharma et al., 2022; Fu et al., 2022). Recent studies also leverage generative models to create semantic adversarial alterations in images (Song et al., 2018; Gowal et al., 2020; Ibrahim et al., 2022; Christensen et al., 2022; Chen et al., 2023). Similarly, Prabhu et al. (2023) has utilized image editing method Hertz et al. (2022b) in order to generate counterfactual examples to evaluate the robustness of vision models.

## 3 BACKBENCH

Our primary goal is to introduce a diverse set of background alterations while ensuring the object's visual presentation and semantic meaning remain intact. This is distinct from prior studies, which either assessed models using artificially generated datasets, modifying both object traits and backgrounds, or employed real images with constrained distribution shifts.

We propose an approach that combines the generative power of diffusion models with the generalization capabilities of foundational models such as SAM (Kirillov et al., 2023) and BLIP-2 (Li et al., 2023a). We achieve this by exploiting the complementary strength of foundational models; image-to-segment, and image-to-text to guide object-preserving diffusion inference for natural and adversarial background-to-object context variations (see Figure 2). Our automated approach effectively generates different datasets under varying distribution shifts, which can be used for benchmarking vision and vision language models. In Section 3.1, we provide preliminaries of diffusion models and vision and language foundational models that serve as the basis for our work. In Section 3.2, we introduce our approach and explain its working in detail for generating diverse background changes.

### 3.1 PRELIMINARIES

**Diffusion Models.** Diffusion models have made remarkable advancements compared to Generative Adversarial Networks (GANs) in the realm of creating realistic images and refining them based on textual guidance. During training, diverse noisy versions $\mathcal{I}_t$ of the clean image $\mathcal{I}$ are fed into the diffusion model $\epsilon_\theta$ at different time steps $t$. The objective for the model is to learn the specific noise added at each time step. The training process of diffusion models comprises of two stages; in the forward process *(first stage)* gaussian noise sampled from a normal distribution $\mathcal{N}(0, I)$ is gradually added to image $\mathcal{I}$ according to a variance schedule $(\beta_t : t = 1, ..., T)$. Using the reparameterization

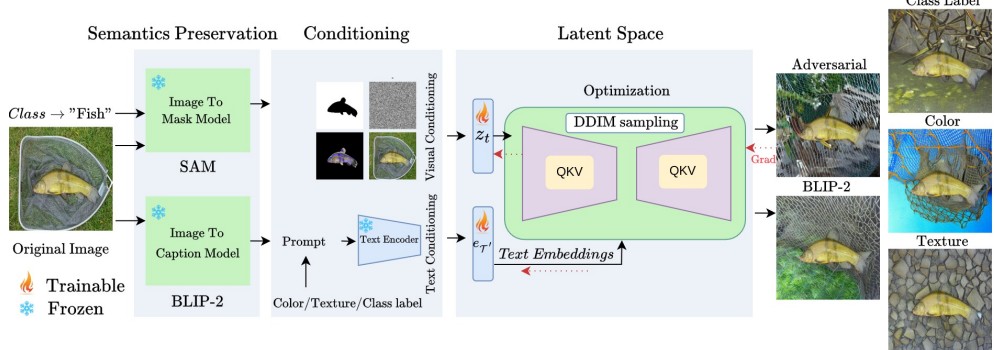

Figure 2: We utilize a stable diffusion inpaint pipeline to generate the counterfactual background of an image. The object mask is obtained from a segmentation model (SAM) by providing the class label as an input prompt. This segmentation mask, along with the original image caption (generated via BLIP-2) is then processed through the diffusion model. In the case of an ensemble attack, both the latent and conditional embeddings at intermediate timesteps are optimized using a loss function.

trick, we can get the noisy image $\mathcal{I}_t$ at any time step as follows:

$$\mathcal{I}_t = \sqrt{\bar{\alpha}_t}\mathcal{I} + \sqrt{1 - \bar{\alpha}_t}\epsilon \quad \epsilon \sim \mathcal{N}(0, I) \tag{1}$$

Here, $\alpha_t = 1 - \beta_t$ and $\bar{\alpha}_t = \prod_{s=1}^{t} \alpha_s$. As $T \to \infty$, $\bar{\alpha}_T \to 0$, which implies $\mathcal{I}_T \sim \mathcal{N}(0, I)$, all the information of the original image $\mathcal{I}$ will be lost. Typically, diffusion models are conditioned on the time step $t$ and additional factors like the class label $y$ or textual description $\mathcal{T}$. However, recent works have extended this conditioning to also consider the clean image $\mathcal{I}$ and its corresponding mask, enabling specialized image editing tasks (Rombach et al., 2022; Saharia et al., 2022). Next, in the reverse process *(second stage)* based on the conditioning, a model $\epsilon_\theta$ is learned to approximate the gaussian parameters at each time step $t$ for the true reverse conditional distribution. Essentially the training objective is to minimize the error between the estimated and actual noise added to the image at different time steps $t$.

$$L_t = ||\epsilon - \epsilon_\theta^t(\mathcal{I}_t, e_\mathcal{T}, \psi)||^2 \tag{2}$$

$e_\mathcal{T}$ is the embedding of conditional guidance through either class label or caption of the image and $\psi$ is any additional conditioning, such as mask or layout of the scene. We use a pre-trained inpainting pipeline from Stable Diffusion (Rombach et al., 2022), specializing in filling masked regions in images using textual instructions. When presented with an image $\mathcal{I}$ and its corresponding mask $\mathcal{M}$, the model utilizes textual conditioning $\mathcal{T}$ to populate the masked region with visual information.

**Foundational Models.** BLIP-2 (Li et al., 2023a) presents an efficient vision-language pre-training approach that utilizes a lightweight Querying Transformer (QFormer) to bridge the modality gap between pre-trained vision and large language models (LLMs). This framework initially passes the image through a pre-trained vision encoder and extracts relevant features via the QFormer. This information is then passed to the pre-trained LLM to obtain a descriptive caption of the image.

Kirillov et al. (2023) present the Segment Anything Model (SAM), that undergoes pre-training on an extensive dataset of high quality images. SAM employs prompts, which can manifest in various forms such as point sets, boxes, masks, or textual input, to demarcate objects within an image. The image initially undergoes an encoding process through a large transformer-based image encoder. Subsequently, both the features extracted from the image and the embeddings of the prompt from a prompt encoder traverse a lightweight decoder, yielding the desired segmentation mask.

In our work, we use BLIP-2 and SAM to condition the diffusion model. BLIP-2 provides textual guidance by extracting a descriptive text $\mathcal{T}$ of the scene in the image, while SAM provides the mask $\mathcal{M}$ that delineates objects from backgrounds.

## 3.2 Changing Object to Background Composition

**Preserving Object Semantics.** In order to modify image backgrounds without affecting the object, an obvious first step would be accurately delineating the object from its background. Additionally, incorporating semantic information in the form of textual descriptions can provide valuable insights into the composition of the scene and guide subsequent modifications.

To achieve this goal, we propose an Object-to-Background Conditioning Module denoted as $\mathcal{C}$, which takes the input image $\mathcal{I}$ and the provided label $\boldsymbol{y}$ as inputs, and returns both the textual prompt $\mathcal{T}$ describing the scene and mask $\mathcal{M}$ encapsulates the object in the image:

$$\mathcal{C}(\mathcal{I}, y) = \mathcal{T}, \mathcal{M} \tag{3}$$

Our conditioning module leverages a promptable segmentation model called SAM (Kirillov et al., 2023) denoted by $\mathcal{S}$. By passing the class information $\boldsymbol{y}$ and the image $\mathcal{I}$ to the model $\mathcal{S}(\mathcal{I}, \boldsymbol{y})$, we obtain the object mask $\mathcal{M}$. Simultaneously, to acquire a description for the image scene, we utilize BLIP-2 (Li et al., 2023b), an image-to-text model denoted as $\mathcal{B}$ to get the necessary prompt $\mathcal{T}$ describing the scene, thereby providing object-to-background context information.

$$\mathcal{B}(\mathcal{I}) = \mathcal{T} \quad ; \quad \mathcal{S}(\mathcal{I}, \boldsymbol{y}) = \mathcal{M} \tag{4}$$

We add the desired background changes by modifying the textual description $\mathcal{T}$ to get $\mathcal{T}'$ (see Appendix A.1). The mask $\mathcal{M}$ and the textual prompt $\mathcal{T}'$ serve as conditioning inputs for the subsequent stage, where we employ a diffusion model to generate diverse background variations. This methodical integration of segmentation and language comprehension offers a fine-grained control over image backgrounds while upholding object semantics, leading to refined object-centric image manipulations.

**Visual and Textual Guidance for Background Modification.** Once we've obtained both visual and textual information $(\mathcal{T}', \mathcal{M})$ from our conditioning module, we leverage the generative power of diffusion models to govern image manipulations. Specifically, we employ a diffusion model that has been trained for inpainting tasks, which has additional conditioning $\psi$ comprising of the image $\mathcal{I}$ and its corresponding mask $\mathcal{M}$. The denoising operation takes place in the latent space instead of the image pixel space, which is facilitated through the use of a variational autoencoder that provides the mapping between images and their respective latent representations. During the inference stage, starting with a standard normal Gaussian noise latent $z_t$, the diffusion model calculates the estimated noise $\hat{\epsilon}_\theta^t$ to be removed from the latent at time step $t$ using a linear combination of the noise estimate conditioned on the textual description $\epsilon_\theta^t(z_t, e_{\mathcal{T}'}, i, m)$ and the unconditioned estimate $\epsilon_\theta^t(z_t, i, m)$:

$$\hat{\epsilon}_\theta^t(z_t, e_{\mathcal{T}'}, i, m) = \epsilon_\theta^t(z_t, i, m) + \lambda \left( \epsilon_\theta(z_t, e_{\mathcal{T}'}, i, m) - \epsilon_\theta^t(z_t, i, m) \right) \tag{5}$$

Here, $(i, m)$ represents the representation of the original image $\mathcal{I}$ and its corresponding mask $\mathcal{M}$ in the latent space. The guidance scale $\lambda$ determines how much the unconditional noise estimate $\epsilon_\theta(z_t, i, m)$ should be adjusted in the direction of the conditional estimate $\epsilon_\theta(z_t, e_{\mathcal{T}'}, i, m)$ to closely align with the provided textual description $\mathcal{T}'$. In this whole denoising process, the mask $\mathcal{M}$ generated from our conditioning module guides the image alterations to the background of the object, while as the textual description $\mathcal{T}'$ which contains information of the original object and the desired background change. This provides additional constraints during the denoising process as to what kind of background should be added while maintaining the faithfulness of the original object.

Our approach allows optimizing the conditioned visual and textual latents $z_t$ and $e_{\mathcal{T}'}$ using a discriminative model $\mathcal{F}_\phi$ to generate adversarial backgrounds.

**Adversarial Background Generation.** For generating adversarial examples the goal of the attacker is to craft perturbations $\delta$ that when added to clean image $\mathcal{I}$ with class label $\boldsymbol{y}$, result in an adversarial image $\mathcal{I}_{adv} = \mathcal{I} + \delta$ which elicits an incorrect response from a classifier model $\mathcal{F}_\phi$ i.e., $\mathcal{F}_\phi(\mathcal{I}_{adv}) \neq \boldsymbol{y}$, where $\phi$ are the model parameters. Usually in pixel-based perturbations, $\delta$ is bounded by a norm-distance, such as $l_2$ or $l_\infty$ norm in order to put a constraint on pixel level changes done to preserve the semantics of the image. However, in our setting the control on the amount of perturbation added is governed by the textual and visual latents passed to the diffsuion model. In our approach (see

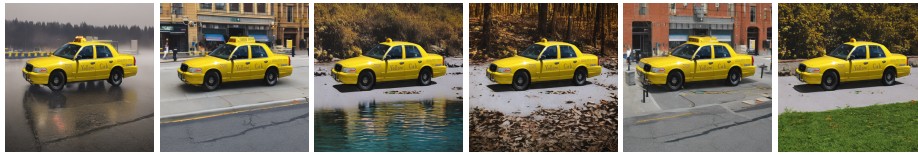

Figure 3: Background diversity achieved using different prompts in each instance. More visual illustrations are provided in Appendix A.3.

Algo. 1), we use the discriminative model $\mathcal{F}_\phi$ to guide the diffusion model $\epsilon_\theta$ to generate adversarial examples by optimizing its latent representations $z_t$ and $e_{\mathcal{T}'}$:

$$\max_{z_t, e_{\mathcal{T}'}} \mathcal{L}_{adv} = \mathcal{L}_{CE}(\mathcal{F}_\phi(\mathcal{I}_{adv}), \boldsymbol{y}) \tag{6}$$

where $\mathcal{L}_{CE}$ is the cross-entropy loss, $e_{\mathcal{T}'}$ is textual embedding and $z_t$ is the denoised latent at time step $t$. $\mathcal{I}_{adv}$ represents the image generated by the diffusion model after it has been denoised using DDIM (Song et al., 2020), a determinitic sampling process in which the the latent update is formulated as:

$$z_{t-1} = \sqrt{\bar{\alpha}_{t-1}}\left(\frac{z_t - \sqrt{1 - \bar{\alpha}_t}\hat{\epsilon}_\theta^t}{\sqrt{\bar{\alpha}_t}}\right) + \sqrt{1 - \bar{\alpha}_{t-1}}\hat{\epsilon}_\theta^t, \quad t = T, \ldots, t-1, \ldots, 1 \tag{7}$$

Our proposed unconstrained adversarial objective $\mathcal{L}_{adv}$ would lead to unrestricted changes in the image background while preserving the object semantics using the mask conditioning from $\mathcal{S}$.

## 4 EXPERIMENTAL PROTOCOLS

**Dataset Preparation.** For classification, we initially gathered 30k images from the ImageNet validation set (Deng et al., 2009), which are correctly classified with high success rate using an ensemble of models; ViT-T, ViT-S (Dosovitskiy et al., 2020), Res-50, Res-152 (He et al., 2016), DenseNet-161 (Huang et al., 2017), Swin-T, and Swin-S (Liu et al., 2021). In order to create a high-quality dataset for our object-to-context variation task, we remove image samples where the boundary between foreground and background is not distinct, e.g., "mountain tent" where the mountain might appear in the background of the tent. This processing results in 15k images. Then for foreground semantic preservation, we utilize a compute efficient variant of SAM, known as FastSAM (Zhao et al., 2023) with class labels as prompts to generate segmentation masks of the foreground object. However, FastSAM encounter challenges in accurately segmenting objects in all images. To address this, we selected images where the mask-creation process demonstrated exceptional accuracy and generated a clear separation between the object of interest and its background. This meticulous selection process yield a curated dataset comprising 5,505 images, representing a subset of 582 ImageNet classes. We refer to this dataset as `IN-Nat`. We perform all natural object-to-context variations using these images. More details are presented in Appendix A.12. Note that our background context variation is a result of the diffusion process which can be computationally expensive for adversarial background optimization. Therefore, for the adversarial backgrounds, we selected a subset of 1,000 images from 500 classes of `IN-Nat` by sampling two images from each class. We refer to this dataset as `IN-Adv`. For object detection, we carefully filtered 1,127 images manually from the COCO 2017 validation set (Lin et al., 2015), with a clear distinction between foreground objects and their background. We refer to these images as `COCO-DC`. We use `COCO-DC` to evaluate both detection and classification. The dataset can have multiple objects in the image. To use this dataset for classification we train the above-mentioned models on the COCO train dataset using labels associated with the object that occupies the highest mask region in the image and evaluated on our generated dataset.

**Diffusion Inference.** We use the pre-trained Inpaint Stable Diffusion v2 model(Rombach et al., 2022) and set the guidance parameter $\lambda$ to 7.5, and use the DDIM sampling (Song et al., 2020) with $T = 20$ timesteps. We craft adversarial examples on `IN-Nat` using Res-50 as the classifier model and maximize the adversarial loss $\mathcal{L}_{adv}$ shown in Eq.6 for 30 iterations. For `COCO-DC`, we maximize the loss in the feature space of the model. Both the text embedding $e_{\mathcal{T}}$ of the prompt

Table 1: Resilience of Transformer and CNN models trained on ImageNet and COCO training sets against our proposed object-to-background context variations. We report top-1 (%) accuracy. We observe that CNN based models are relatively more robust than Transformers.

| Datasets | Background | Transformers | | | | CNN | | | |
|---|---|---|---|---|---|---|---|---|---|
| | | ViT-T | ViT-S | Swin-T | Swin-S | Res-50 | Res-152 | DenseNet-161 | Average |
| IN-Nat | Original | 96.04 | 98.18 | 98.65 | 98.84 | 98.65 | 99.27 | 98.09 | 98.25 |
| | Class label | 92.82 | 94.75 | 96.18 | 96.55 | 97.24 | 97.56 | 95.8 | 95.84$_{(-2.41)}$ |
| | BLIP-2 Caption | 86.77 | 90.41 | 92.71 | 93.60 | 94.46 | 95.35 | 91.62 | 92.13$_{(-6.12)}$ |
| | Color | 70.64 | 84.52 | 86.84 | 88.84 | 89.44 | 92.89 | 83.19 | 85.19$_{(-13.06)}$ |
| | Texture | 68.24 | 79.73 | 81.09 | 84.41 | 83.21 | 87.66 | 77.29 | 80.23$_{(-18.02)}$ |
| IN-Adv | Original | 95.01 | 97.50 | 97.90 | 98.30 | 98.50 | 99.10 | 97.20 | 97.64 |
| | Adversarial | 18.40 | 32.10 | 25.00 | 31.70 | 2.00 | 28.00 | 14.40 | 21.65$_{(-75.99)}$ |
| COCO-DC | Original | 82.96 | 86.24 | 88.55 | 90.23 | 88.55 | 89.08 | 86.77 | 87.21 |
| | BLIP-2 Caption | 82.69 | 84.73 | 86.24 | 86.95 | 88.46 | 86.69 | 85.01 | 85.67$_{(-1.54)}$ |
| | Color | 55.54 | 61.04 | 70.09 | 72.13 | 74.97 | 75.10 | 66.19 | 66.66$_{(-20.55)}$ |
| | Texture | 52.52 | 58.82 | 68.05 | 70.09 | 70.71 | 74.77 | 63.79 | 63.99$_{(-23.22)}$ |
| | Adversarial | 49.68 | 55.72 | 61.93 | 69.12 | 55.45 | 61.13 | 57.76 | 58.68$_{(-28.52)}$ |

$\mathcal{T}$ (initialized with BLIP-2) and denoised latent $z_t$ are optimized from denoising time step $t = 4$ using AdamW (Loshchilov & Hutter, 2017) with a learning rate of $0.1$. All experiments were conducted using a single NVIDIA-A100 GPU.

**Vision Models.** *a) Natural ImageNet Training:* We evaluate seven naturally ImageNet-trained vision transformers and convolutional neural networks (CNNs). Specifically we use ViT-T, ViT-S (Dosovitskiy et al., 2020), Res-50, Res-152 (He et al., 2016), DenseNet-161 (Huang et al., 2017), Swin-T, and Swin-S (Liu et al., 2021). *b) Adversarial ImageNet Training.* We also evaluate adversarial ImageNet-trained models including ResAdv-18, ResAdv-50, and WideResAdv-50 at various perturbation budget of $\ell_\infty$ and $\ell_2$ (Salman et al., 2020). *c) Stylized ImageNet Training.* We evaluate the DeiT-T and DeiT-S models trained on a stylized version of the ImageNet dataset (Naseer et al., 2021; Geirhos et al., 2018). *Multimodal Training.* Additionally, we explored seven vision language foundational models within CLIP (Radford et al., 2021) and EVA-CLIP (Sun et al., 2023).*d) Segmentation and Detection.* We evaluate Mask-RCNN for segmentation and object detection respectively using our proposed background-to-object variations. Further evaluations on FastSAM (Zhao et al., 2023) and DETR (Carion et al., 2020) are reported in Appendix A.6 and A.5. *e) Image Captioning.* Further, we evaluate the robustness of a recent image captioning model BLIP-2 (Li et al., 2023b), using our generated dataset.

**Evaluation Metrics:** We report results using the top-1 accuracy (%), Intersection Over Union (IoU), Average Precision(AP) and Recall(AR), and CLIP text similarity score for classification, segmentation, object detection, and captioning tasks, respectively.

**Text Prompts Conditioning.** During inducing background variations, we use the following text prompts templates; Class Label: "*A picture of a class*" where *class* is the class name of the image, Caption: "*captions from BLIP-2 model*, Color: "*A picture of ___ background*" where __ is replaced with red, green, blue, and colorful, Texture: "*A picture of __ background* where __ is replaced with textured, intricately rich textures, colorful textures, distorted textures, Adversarial: "*captions from BLIP-2 model*". Note that for adversarial setting the text prompts get updated after optimization. We observe similar trends across different color and texture prompts and report the worst-performing one. Detailed analysis across different color and texture prompts is provided in Appendix A.7.

## 4.1 RESULTS

**Natural ImageNet Training.** Our evaluation demonstrates a consistent decline in accuracy for both transformer-based and CNN models when exposed to diverse object-to-background changes. This decrease is especially noticeable in texture and color backgrounds across both IN-Nat and COCO-DC, as summarized in Table 1. We found that as we moved from purely transformer-based architectures to convolution-based architectures, there was an overall improvement in accuracy across different background changes. For instance the average accuracy across all backgrounds for ViT-T, Swin-T, and Res-50 on IN-Nat is 79.61%, 89.20% and 91.08% respectively. A similar trend can be seen on COCO-DC dataset. Further, we observe that as the model capacity is increased across

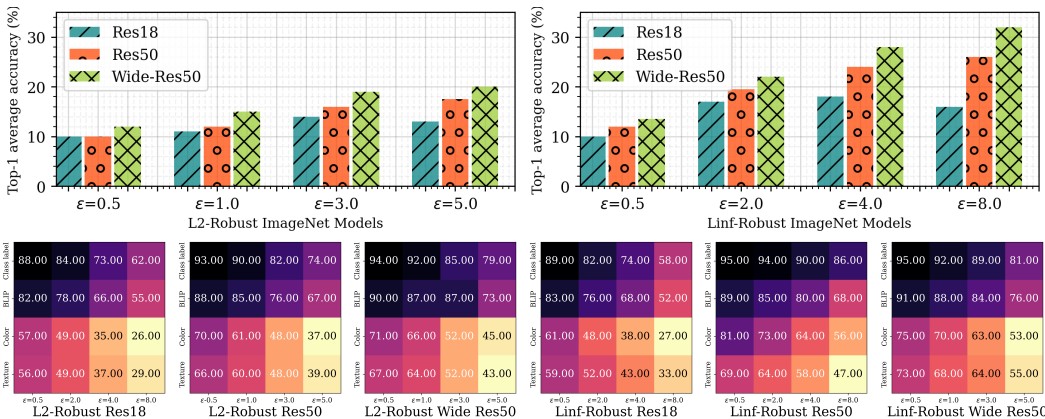

Figure 4: Adversarially trained models performance on `IN-Nat` and `In-Adv`. The top row plots the Top-1(%) accuracy achieved by adversarially trained ResNet models on adversarial background changes and the bottom row indicates for the case of non-adversarial background changes.

Table 2: Comparative Evaluation of Zero-shot CLIP and Eva CLIP Vision-Language Models on `IN-Nat` and `IN-Adv`. Top-1(%) accuracy is reported. We find that Eva CLIP models showed more robustness in all object-to-background variations.

| Datasets | Background | CLIP | | | | | | | |
|---|---|---|---|---|---|---|---|---|---|
| | | ViT-B/32 | ViT-B/16 | ViT-L/14 | Res50 | Res101 | Res50x4 | Res50x16 | Average |
| IN-Nat | Original | 75.56 | 81.56 | 88.61 | 73.06 | 73.95 | 77.87 | 83.25 | 79.12 |
| | Class label | 80.83 | 84.41 | 89.41 | 78.87 | 79.33 | 81.94 | 85.67 | 82.92(+3.80) |
| | BLIP-2 Captions | 69.33 | 73.66 | 79.07 | 67.44 | 68.70 | 71.55 | 75.78 | 72.22(-6.90) |
| | Color | 53.02 | 63.08 | 71.42 | 53.53 | 55.87 | 60.05 | 71.28 | 61.18(-17.94) |
| | Texture | 51.01 | 62.25 | 69.08 | 51.35 | 53.46 | 61.10 | 70.33 | 59.79(-19.33) |
| IN-Adv | Original | 73.90 | 79.40 | 87.79 | 70.69 | 71.80 | 76.29 | 82.19 | 77.43 |
| | Adversarial | 25.5 | 34.89 | 48.19 | 18.29 | 24.40 | 30.29 | 48.49 | 32.87(-46.25) |
| Datasets | Background | EVA-CLIP | | | | | | | |
| | | g/14 | g/14+ | B/16 | L/14 | L/14+ | E/14 | E/14+ | Average |
| IN-Nat | Original | 90.80 | 93.71 | 90.24 | 93.71 | 93.69 | 95.38 | 95.84 | 93.34 |
| | Class label | 90.48 | 93.53 | 90.20 | 93.47 | 93.49 | 94.78 | 95.18 | 93.02(-0.32) |
| | BLIP-2 Caption | 80.56 | 85.23 | 81.88 | 85.28 | 86.24 | 88.13 | 88.68 | 85.14(-8.20) |
| | Color | 77.25 | 83.96 | 76.24 | 83.63 | 85.79 | 88.70 | 88.33 | 83.41(-9.93) |
| | Texture | 75.93 | 82.76 | 74.44 | 82.56 | 86.35 | 87.84 | 88.44 | 82.62(-10.72) |
| IN-Adv | Original | 88.80 | 92.69 | 89.19 | 91.10 | 91.99 | 93.80 | 94.60 | 91.74 |
| | Adversarial | 55.59 | 62.49 | 48.70 | 65.39 | 73.59 | 70.29 | 73.29 | 64.19(-27.55) |

different model families, the robustness to background changes also increases. As is evident, the models are most vulnerable to adversarial background changes, resulting in a significant drop in average accuracy. In Figure 6 the same effect is shown by visualizing the loss surface of a classifier (ViT-S) across different background changes. We provide extensive visualizations and results on different background prompts in Appendix A.3 and A.7.

**Adversarial ImageNet Training.** As can be seen from the Figure 4 *(bottom row)*, adversarial trained models show a significant drop in accuracy on `IN-Nat` with object-to-background changes, implying that the robustness gained by these models does not transfer to different distribution shifts. However, when we evaluate these models on adversarial examples crafted on `IN-Adv`, the performance improves with an increase in adversarial robustness($\epsilon$) of the models (see Figure 4 *(top row)*). We also observe models with more capacity perform better, similar to results on natural training.

**Multimodal Training.** We observe that the CLIP the zero-shot robustness on different background variations decreases similar to results mentioned in Table 1. However, for background variations induced using class label information the performance increases. This could be because of the CLIP text encoder utilized for textual conditioning of the diffusion model. On EVA-CLIP, which proposed changes to stabilize the training of CLIP models on large-scale datasets, we observe significant

Table 3: Stylized Training Evaluation

| Datasets | Background | Stylized Trained models | | |
|---|---|---|---|---|
| | | DeiT-S | DeiT-T | Average |
| IN-Nat | Original | 91.22 | 87.21 | 89.21 |
| | Class label | 89.35 | 85.35 | 87.35 (-1.86) |
| | BLIP-2 Caption | 84.01 | 79.19 | 81.60 (-7.61) |
| | Color | 66.57 | 57.54 | 62.05 (-27.15) |
| | Texture | 64.08 | 54.82 | 59.45 (-29.76) |
| IN-Adv | Original | 89.60 | 85.90 | 87.75 |
| | Adversarial | 15.90 | 10.80 | 13.35 (-74.40) |

Table 4: Image-to-Caption (BLIP-2) Evaluation

| Dataset | Background | CLIP Score |
|---|---|---|
| IN-Nat | Class Label | 0.75 |
| | BLIP-2 Caption | 0.84 |
| | Color | 0.66 |
| | Texture | 0.67 |
| IN-Adv | Adversarial | 0.62 |

Table 5: Mask AP and Segment AP score on COCO-DC

| Background | Box AP | Segment AP |
|---|---|---|
| Original | 57.99 | 56.29 |
| BLIP-2 Caption | 47.40 | 44.75 |
| Color | 48.12 | 45.09 |
| Texture | 45.79 | 43.07 |
| Adversarial | 37.10 | 34.91 |

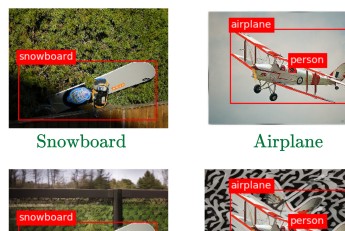 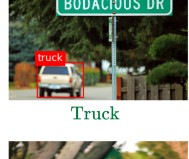 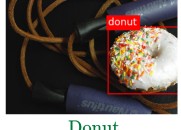 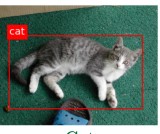

Snowboard    Airplane    Truck    Donut    Cat

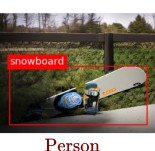 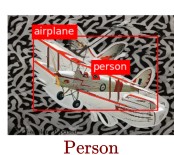 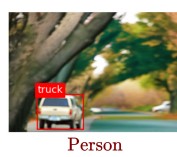 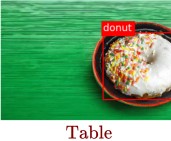 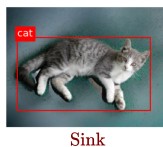

Person    Person    Person    Table    Sink

Figure 5: Correct predictions by Mask-RCNN and Res-50 on the original image *(top row)* and the corresponding predictions on altered backgrounds *(bottom row)*.

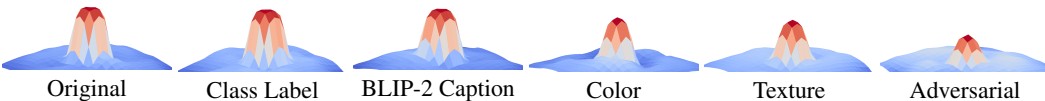

Original    Class Label    BLIP-2 Caption    Color    Texture    Adversarial

Figure 6: The loss surfaces *(flipped)* of the ViT-S depicted on IN-Nat. Significant distribution shifts result in narrow and shallow surfaces at convergence.

improvement in zero-shot performance across all background changes. Further results delving into the comparison between multimodal and unimodal models are provided in Appendix A.7.

**Stylized ImageNet Training.** Stylized ImageNet training (Geirhos et al., 2018), which helps models to focus on the foreground of the scene (Naseer et al., 2021), remains vulnerable to our object-to-background variations on IN-Nat and IN-Adv (Table 3).

**Segmentation and Detection.** We observe that the AP scores on detection and instance segmentation on different background variations decrease compared to the original (see Table 5). The adversarial background results in the lowest AP scores, but still remains at a reasonable level given that the adversarial examples are generated using a classification model, with limited cross-task transferability. Moreover, our qualitative observations suggest detection and segmentation models exhibit greater resilience to changes in the background compared to classifiers (Figure 5 and Appendix A.5).

**Image Captioning.** Table 4 reports the CLIP score between the captions generated on the clean and generated images using BLIP-2 model. The scores decrease across color, texture and adversarial background changes.

## 5 CONCLUSION

In this study, we propose a new benchmark for evaluating the resilience of current vision and vision-language models to object-to-background context on real images. Our proposed framework, Back-Bench, utilizes the capabilities of image-to-text and image-to-segmentation foundational models to preserve the semantics and appearance of the object while adding diverse background changes in real images through textual guidance of the diffusion model. BackBench offers a complimentary evaluation protocol to the existing ones in the literature. We anticipate this will pave the way for a more thorough evaluation of vision models, consequently driving the development of effective methods for improving their resilience.

**Reproducibility Statement**: Our method uses already available pre-trained models and the codebase is based on several open source implementations. We highlight the main components used in our framework for reproducing the results presented in our paper, **a) Diffusion Inpainting Implementation:** We use the open-source implementation of Stable-Diffusion-Inpainting method (`https://github.com/huggingface/diffusers/blob/main/src/diffusers/`) with available pretrained weights *(Stable-Diffusion-v-1-2)* for background generation. **b) Image-to-Segment Implementation:** We use the official open-source implementation of FastSAM (`https://github.com/CASIA-IVA-Lab/FastSAM`) to get the segmentation masks of filtered ImageNet dataset. **c) Image-to-Text Implementation:** We use the official open-source implementation of BLIP-2(`https://github.com/salesforce/LAVIS/tree/main/projects/blip2`) to get the captions for each image. We will also provide captions for each image in our dataset. **d) Adversarial Attack:** We intent to open-source our codebase and release the script for crafting adversarial examples. **e) Dataset:** In the paper, we describe the procedure of filtering the images from ImageNet and COCO val. set. Furthermore, we will provide the filtered datasets, object masks as well as prompts used to generate different backgrounds.

**Ethics Statement**: Our work focuses on evaluating resilience of current vision and language models against natural and adversarial background changes in real images. This work can be utilized by an attacker to generate malicious backgrounds on real images as well as generate adversarial backgrounds which can fool the deployed computer-vision systems. Nevertheless, we believe that our research will pave the way for improved evaluation protocols to assess the resilience of existing models. This, in turn, is likely to drive the development of enhanced techniques for bolstering the resilience of deployed systems. Since we are benchmarking vision and vision-language models using a subset of images from publicly available ImageNet and COCO datasets, it's relevant to mention that these datasets are known to have images of people which poses a privacy risk and further it is known to have biases which can encourage social stereotypes. In the future, we intend to benchmark our models on a less biased dataset to mitigate these concerns and ensure a fair evaluation.

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
