## A    APPENDIX

OVERVIEW

### A.1    LIST OF PROMPTS

We provide the list of prompts that are used to guide the diffusion model to generate diverse background changes, encompassing different distribution shifts with respect to the original data distribution.

Table 6: Prompts used to create background alterations

| Background | Prompts |
|---|---|
| Class label | "This is a picture of a *class name*." |
| BLIP-2 Caption | Captions generated from BLIP-2 image to caption |
| $\text{Color}_{\text{prompt-1}}$ | "This is a picture of a vivid red background." |
| $\text{Color}_{\text{prompt-2}}$ | "This is a picture of a vivid green background." |
| $\text{Color}_{\text{prompt-3}}$ | "This is a picture of a vivid blue background." |
| $\text{Color}_{\text{prompt-4}}$ | "This is a picture of a vivid colorful background." |
| $\text{Texture}_{\text{prompt-1}}$ | "This is a picture of textures in the background." |
| $\text{Texture}_{\text{prompt-2}}$ | "This is a picture of intricately textured background." |
| $\text{Texture}_{\text{prompt-3}}$ | "This is a picture of colorful textured background." |
| $\text{Texture}_{\text{prompt-4}}$ | "This is a photo of distorted textures in the background." |
| Adversarial | Captions generated from BLIP-2 image to caption. |

## A.2 ALGORITHM

We provide the algorithm (Algo. 1) for our approach of generating adversarial backgrounds by optimizing the textual and visual conditioning of the diffusion model. We also tried to optimize only the conditional embeddings or the latent embeddings, but achieve better attack success rate by optimizing both. Note that for crafting adversarial examples on COCO-Dc we use ImageNet trained resnet50 classifiers and our adversarial objective is to maximize the feature representation distance between clean and adversarial samples. Furthermore, for introducing desired non-adversarial background changes using the textual description $\mathcal{T}'$, the optimization of the latent and embedding is not needed.

---

**Algorithm 1** Background Generation

---

**Require:** Conditioning module $\mathcal{C}$, Diffusion model $\epsilon_\theta$, Autoencoder $\mathcal{V}$, CLIP text encoder $\psi_{\text{CLIP}}$, image $\mathcal{I}$, class label $\boldsymbol{y}$, classifier $\mathcal{F}_\phi$, denoising steps $T$, guidance scale $\lambda$, attack iterations $N$, and learning rate $\beta$ for AdamW optimizer $\mathcal{A}$.

1: Get the textual and visual conditioning from the image $\mathcal{I}$

$$\mathcal{C}(\mathcal{I}, \boldsymbol{y}) = \mathcal{T}, \mathcal{M}$$

2: Modify $\mathcal{T}$ to $\mathcal{T}'$ for desired background change.
3: Map the mask $\mathcal{M}$ and image $\mathcal{I}$ to latent space: $i, m \leftarrow \mathcal{V}_{\text{ENC}}(\mathcal{I}, \mathcal{M})$
4: Get the embedding of the textual discription $\mathcal{T}'$: $e_{\mathcal{T}'} \leftarrow \psi_{\text{CLIP}}(\mathcal{T}')$
5: Randomly initialize the latent $z_T$
6: Get the denoised latent $z_t$ at time step $t$.
7: **for** $n \in [1, 2, \ldots N]$ **do**
8:     **for** $t \in [t, t+1, \ldots T]$ **do**
9:         $\hat{\epsilon}_\theta^t(z_t, e_{\mathcal{T}'}, i, m) = \epsilon_\theta^t(z_t, i, m) + \lambda\left(\epsilon_\theta(z_t, e_{\mathcal{T}'}, i, m) - \epsilon_\theta^t(z_t, i, m)\right)$
10:         From noise estimate $\hat{\epsilon}_\theta$ get $z_{t-1}$.
11:     **end for**
12:     Project the latents to pixel space: $\mathcal{I}_{adv} \leftarrow \mathcal{V}_{\text{DEC}}(z_0)$
13:     Compute Adversarial Loss:

$$\mathcal{L}_{adv} = \mathcal{L}_{CE}(\mathcal{F}_\phi(\mathcal{I}_{adv}), \boldsymbol{y})$$

14:     Update $z_t$ and $e_{\mathcal{T}}$ using $\mathcal{A}$ to maximize $\mathcal{L}_{adv}$:

$$z_t, e_{\mathcal{T}'} \leftarrow \mathcal{A}\left(\nabla_{z_t}\mathcal{L}_{adv}, \nabla_{e_{\mathcal{T}'}}\mathcal{L}_{adv}\right)$$

15: **end for**
16: 
17: Generate Adversarial image $\mathcal{I}_{adv}$ using updated $z_t$ and $e_{\mathcal{T}'}$.

---

### A.3 DIVERSITY AND DIFFUSION PARAMETER ABLATION

In this section, we qualitatively analyze the diversity in visual results of the diffusion model. In Figure 7, we show that keeping textual and visual guidance fixed, the diffusion model is still able to generate diverse changes with similar background semantics at different seeds for the noise $z_T$. Furthermore, we explore the diversity in genrating realistic background changes across an original image by using diverse class agnostic textual prompts, capturing different realistic backgrounds. Figure 8 and 9 show some of the qualitative results obtained on IN-Nat samples using prompts generated from ChatGPT (See Table 7). Furthermore, we show the visual examples of color, texture, and adversarial attack on IN-Nat dataset in Figure 10, 11, and 12. We also provide a visualization in Figure 13 showing the effect of changing diffusion model parameters.

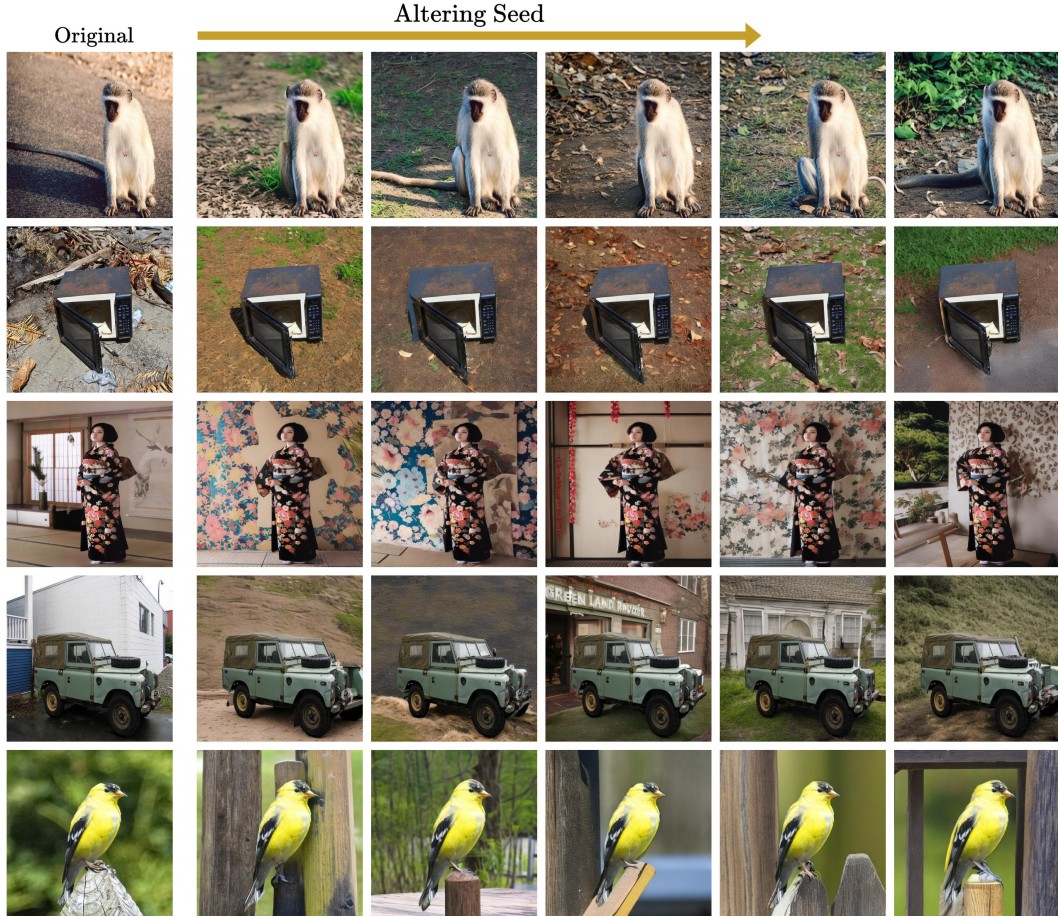

Figure 7: In this figure, examples are generated using BLIP-2 captions by altering the seed from left to right in the row. This highlights the high diversity achievable with the diffusion model when employing different starting noise latents.

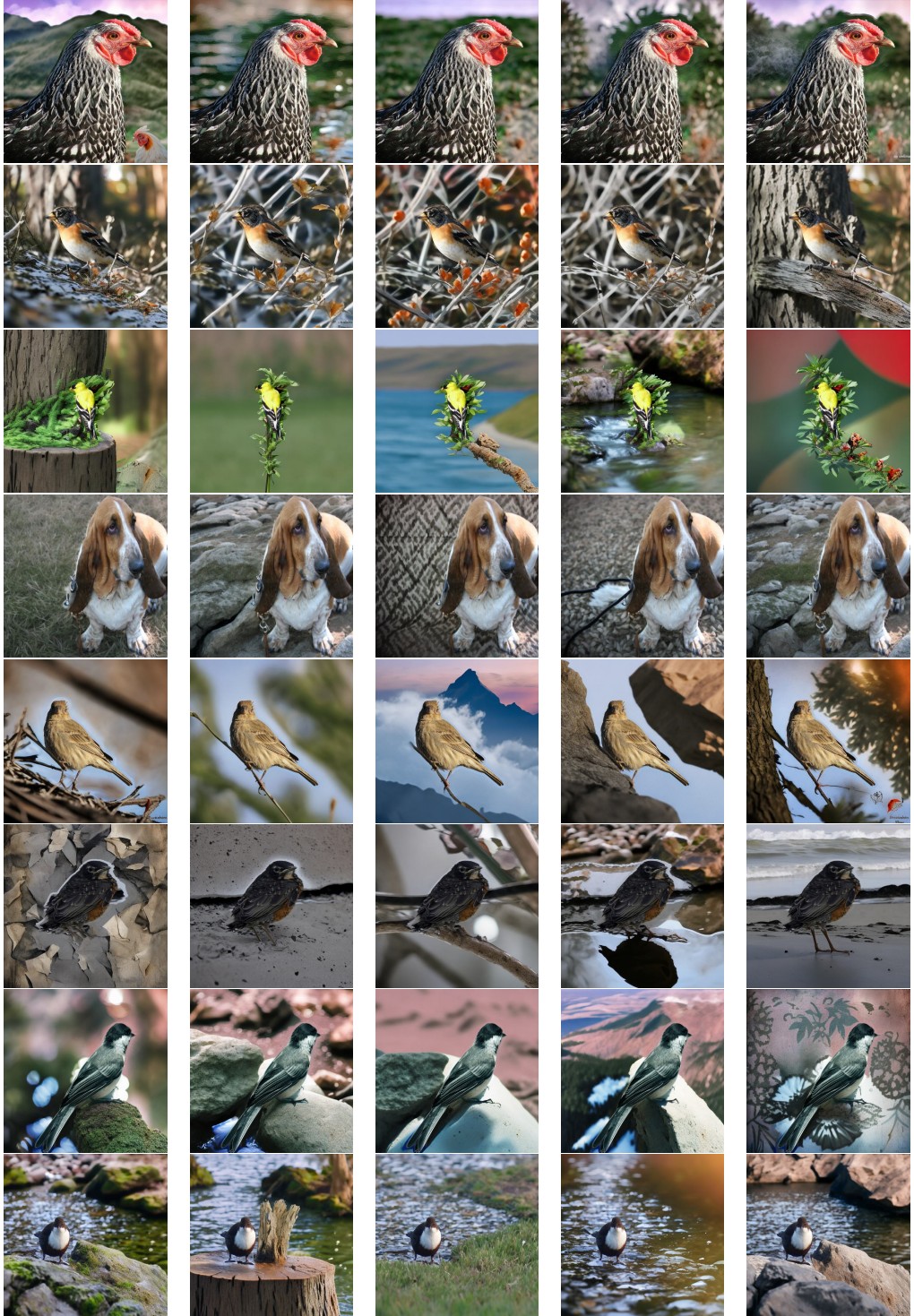

Figure 8: Using diverse prompts to capture for diverse background shifts on samples from `IN-Nat`.

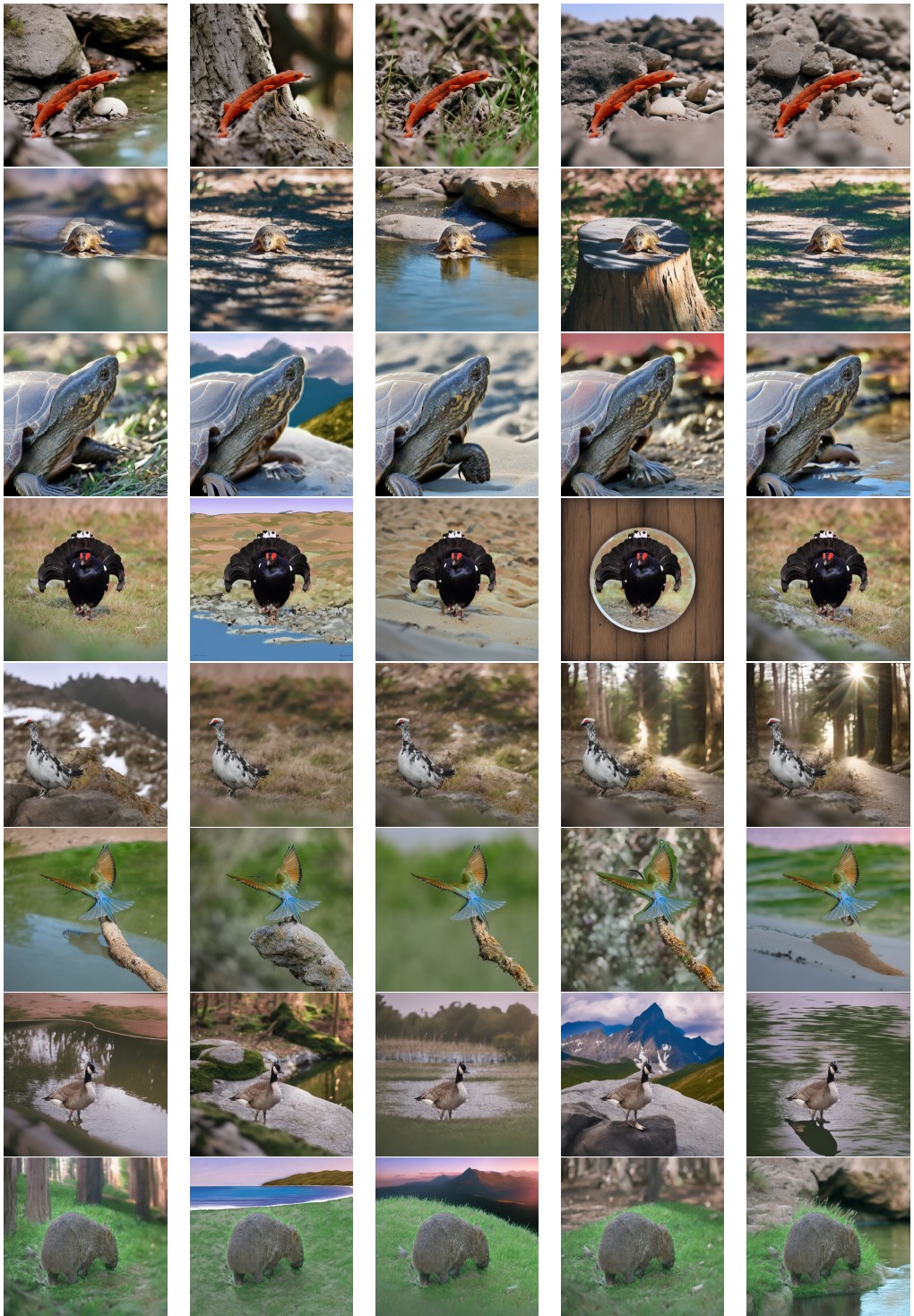

Figure 9: Using diverse prompts to capture for diverse background shifts on samples from `IN-Nat`.

Table 7: ChatGPT prompts used to create diverse and realistic background changes. In the prompts provided the _ is replaced with original class name of the object

- - - - - - - - - - - - - - - - - - - - - - - - - - - - - - - - - - - - - - - - - - - - - - -

*"A realistic photo of _ on a grassy field with a greenish hue background."*
*"A realistic photo of _ on a sandy beach with a bluish tint ocean background."*
*"A realistic photo of _ on a hilltop with a panoramic and slightly reddish background."*
*"A realistic photo of _ on a leafy surface with a vibrant, natural coloration background."*
*"A realistic photo of _ on a forest floor with dappled sunlight and natural hues background."*
*"A realistic photo of _ by a riverbank with a tranquil water stream and bluish tones background."*
*"A realistic photo of _ on a mountain trail with rugged terrain and earthy tones background."*
*"A realistic photo of _ on a mossy rock with a lush, green and slightly yellowish forest background."*
*"A realistic photo of _ on a tree stump with a woodland setting and brownish tones background."*
*"A realistic photo of _ in a forest glade with dappled sunlight background."*
*"A realistic photo of _ on a moss-covered rock with a tranquil forest and greenery background."*
*"A realistic photo of _ on a rocky riverbed with flowing water and natural hues background."*
*"A realistic photo of _ on a forest path with a beautiful, peaceful forest background."*
*"A realistic photo of _ on a beach with a tranquil ocean view and shades of blue background."*
*"A realistic photo of _ on a mountain ledge with a stunning valley view and natural sky background."*
*"A realistic photo of _ by a forest stream with a serene and peaceful natural setting and background."*
*"A realistic photo of _ on a rocky outcrop with a rugged and scenic mountain background."*
*"A realistic photo of _ on a forest floor with a peaceful and natural setting with lush greenery and earthy tones background."*
*"A realistic photo of _ on a sandy beach with a beautiful sunrise and gentle waves rolling onto the shore and warm tones background."*
*"A realistic photo of _ on a mountain ledge with a stunning valley view background."*
*"A realistic photo of _ on a forest path with the sun shining through the trees and warm background."*

- - - - - - - - - - - - - - - - - - - - - - - - - - - - - - - - - - - - - - - - - - - - - - -

*"A photo of _ with a vivid greenish color background."*
*"A photo of _ with a vivid bluish color background."*
*"A photo of _ with a vivid reddish color background."*
*"A photo of _ with a textured background featuring intricate patterns and soft hues."*
*"A photo of _ with a textured background resembling natural stone with earthy tones."*
*"A photo of _ with a textured background resembling wood grain with natural hues."*
*"A photo of _ with a textured background resembling fabric with soft colorations."*
*"A photo of _ with a textured background resembling a painted canvas with artistic tones."*
*"A photo of _ with a textured background resembling crumpled paper with muted hues."*
*"A photo of _ with a textured background resembling sand with natural shades."*
*"A photo of _ with a textured background resembling flowing water with soft tones."*
*"A photo of _ with a textured background resembling a starry night sky with dimmed colors."*
*"A photo of _ with a textured background resembling fluffy clouds with light and airy hues."*

- - - - - - - - - - - - - - - - - - - - - - - - - - - - - - - - - - - - - - - - - - - - - - -

*"A realistic photo of _ with a slightly blurred and diffused background."*
*"A realistic photo of _ with a misty and foggy atmosphere in the background."*
*"A realistic photo of _ with a hazy and ethereal background."*
*"A realistic photo of _ with a softly blurred and dreamy background."*
*"A realistic photo of _ with a gentle and soft-focus background."*
*"A realistic photo of _ with a slightly obscured and blurred background."*
*"A realistic photo of _ with a diffused and unfocused background."*
*"A realistic photo of _ with a veiled and indistinct background."*
*"A realistic photo of _ with a subtly blurred and obscured background."*
*"A realistic photo of _ with a mist-covered and dreamlike background."*

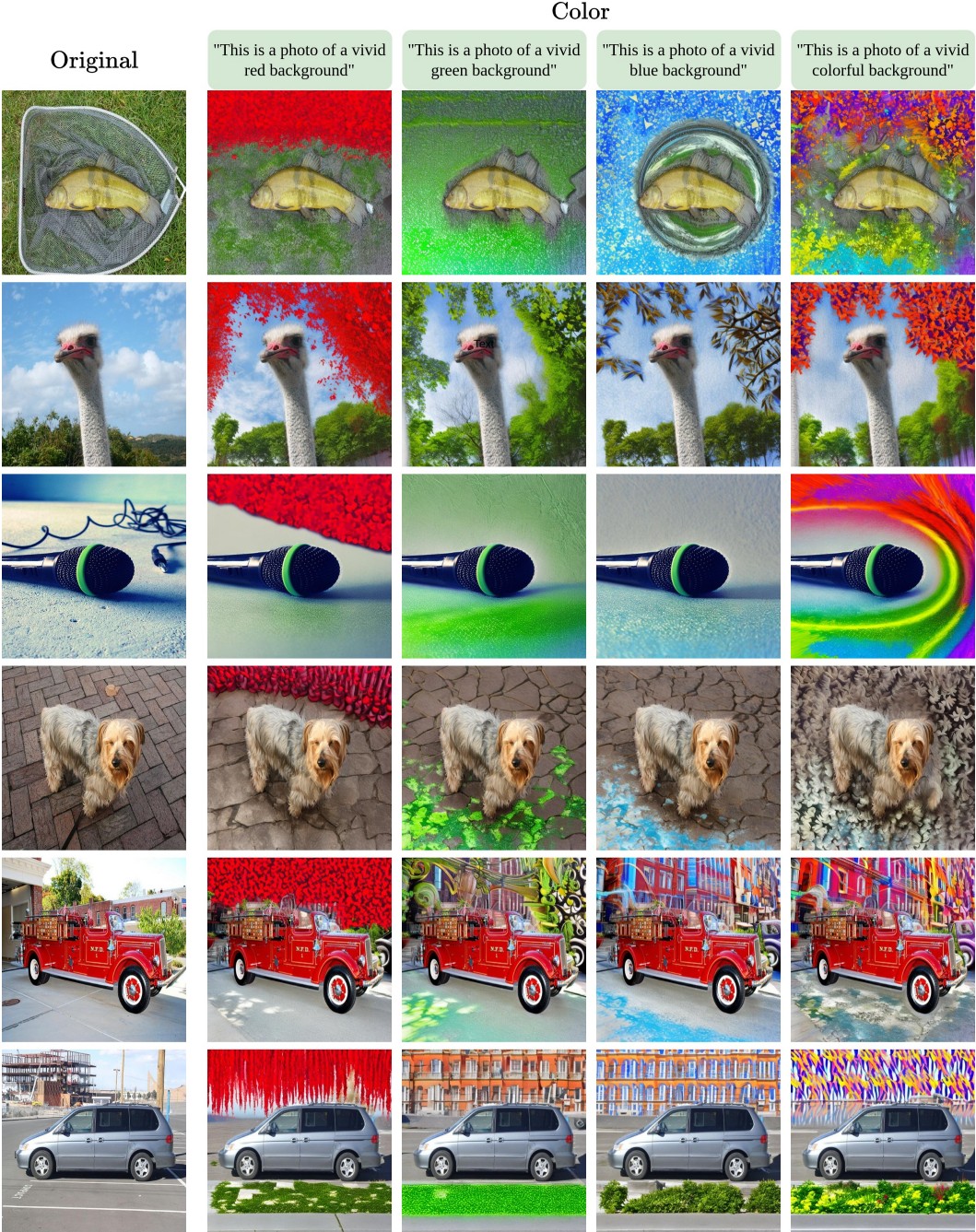

Figure 10: Images generated through diverse color prompts on `IN-Nat`.

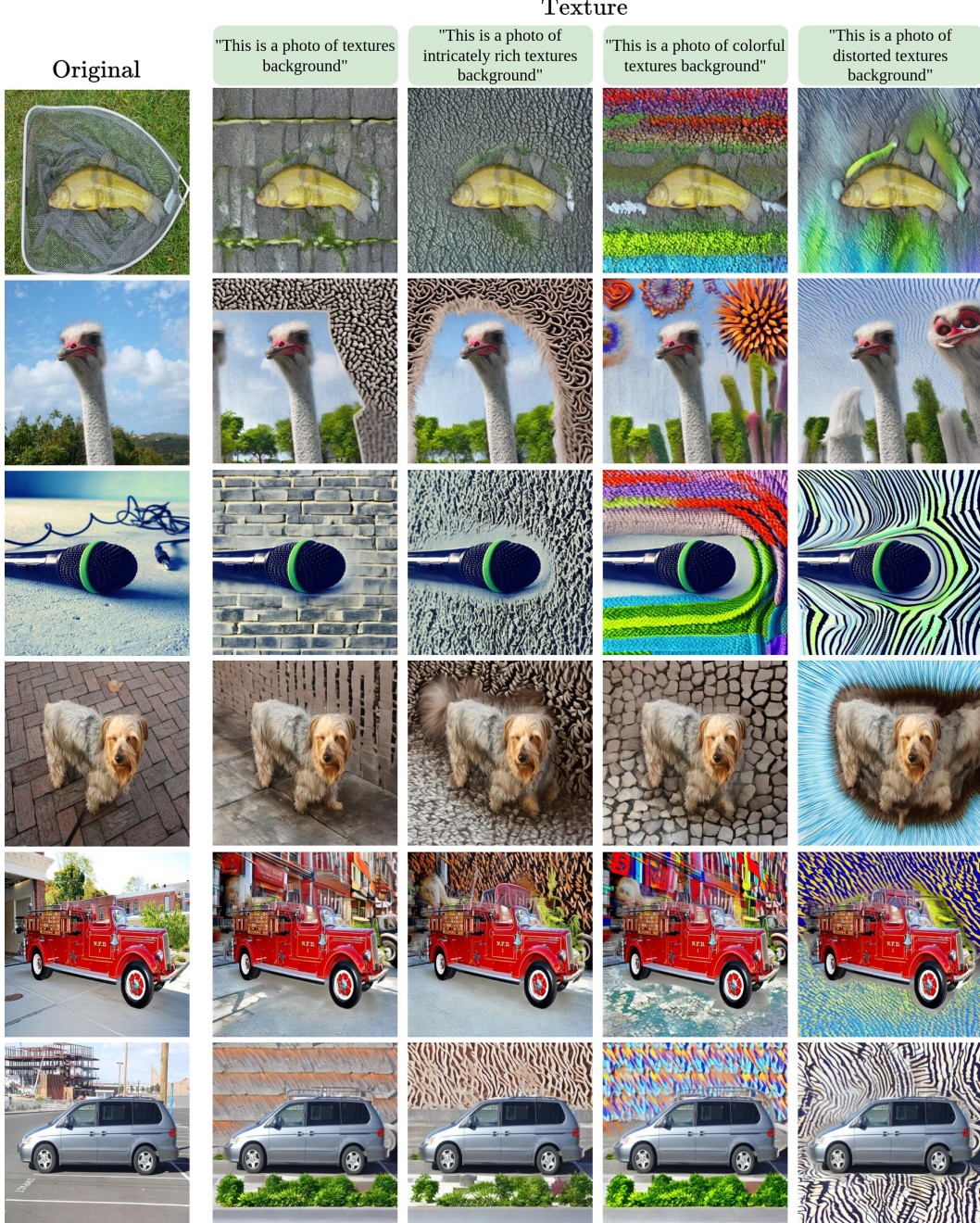

Figure 11: Images generated through diverse texture prompts on `IN-Nat`.

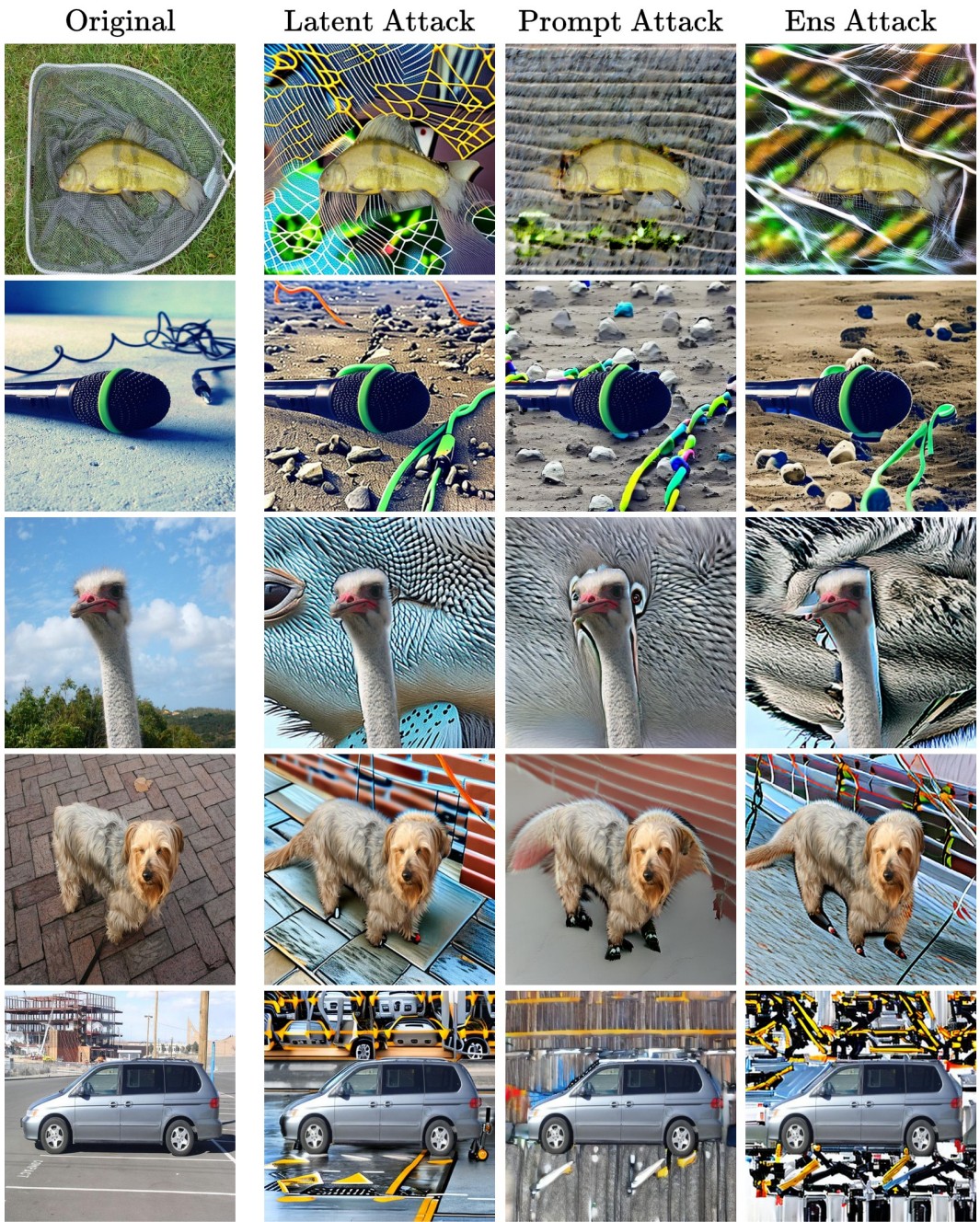

Figure 12: Images generated under various attack scenarios on `IN-Adv`. Here we show the visualization for latent, prompt, and ensemble attack that are generated by optimizing latent, text prompt embeddings, and both respectively.

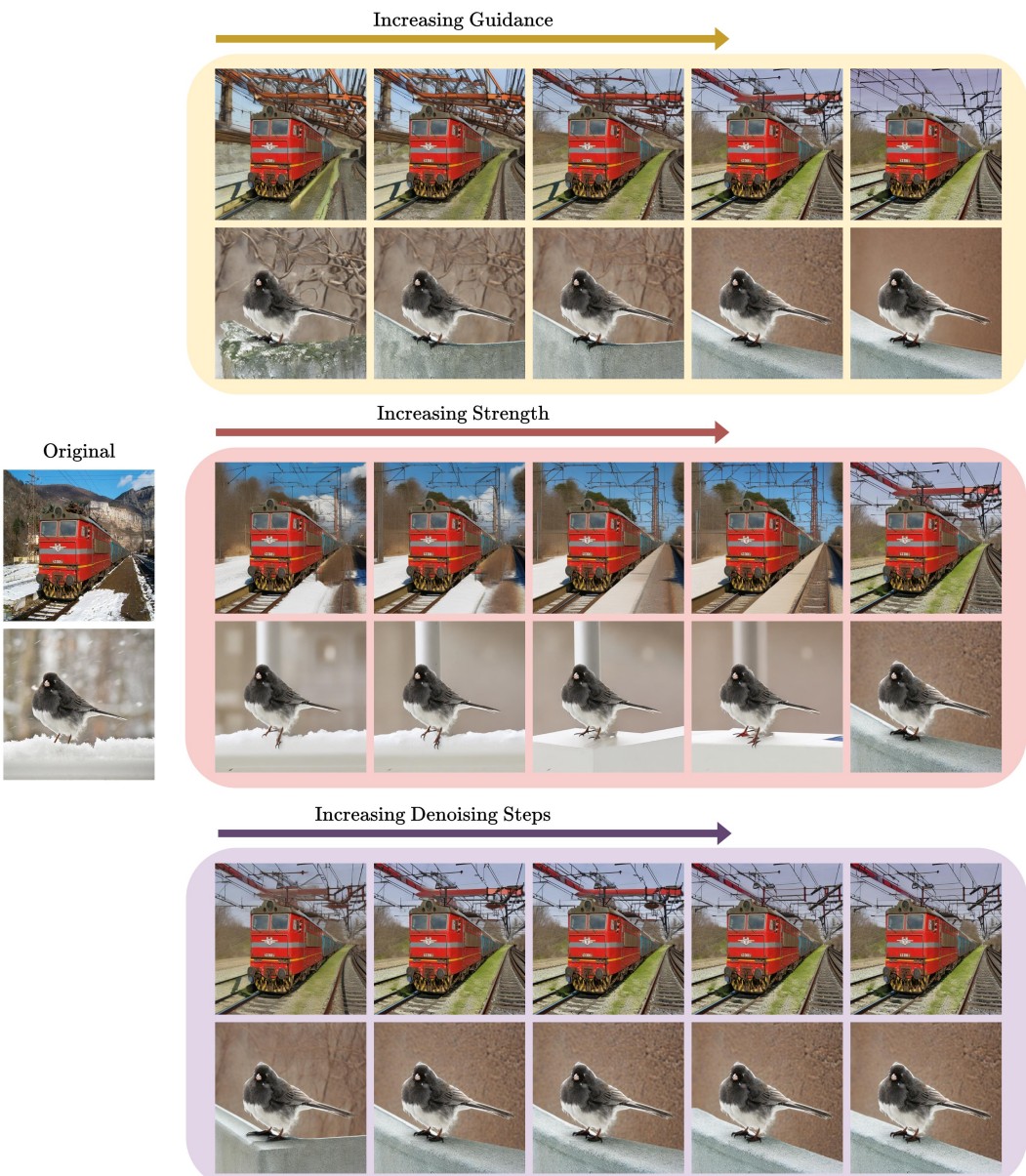

Figure 13: Visualization on samples taken from IN-Nat. Varying parameters like guidance, strength, and denoising steps while using BLIP-2 caption as the prompt. Increasing guidance leads to more fine-detailed background changes. Additionally, greater strength correlated with more pronounced alterations from the original background. And, augmenting diffusion steps improves image quality.

## A.4 Vision Language model for Image Captioning

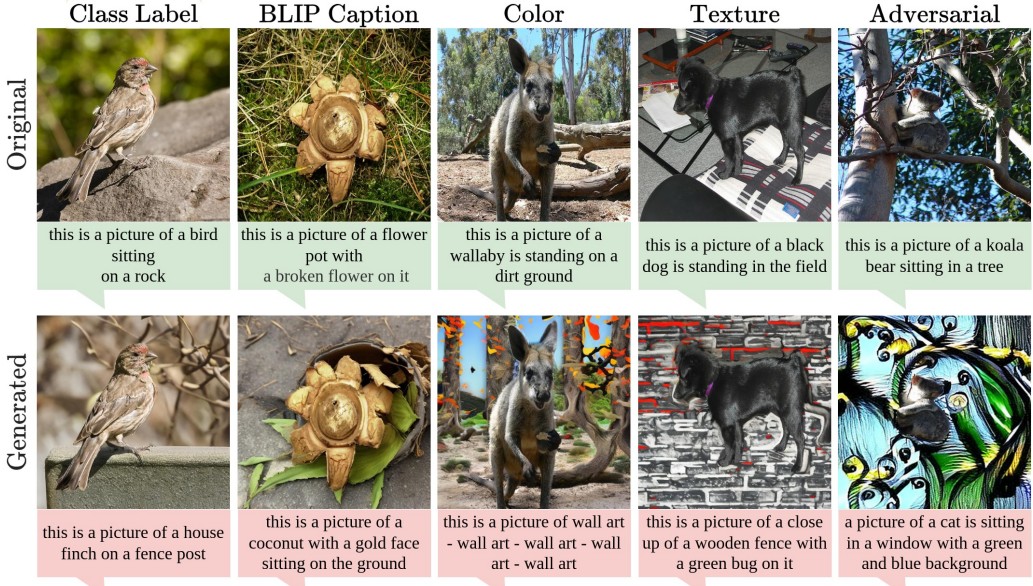

Figure 14: A visual comparison of BLIP-2 captions on clean and generated datasets. The top row shows captions on clean images, while the bottom row displays captions on generated images. As background complexity increases, BLIP-2 fails to accurately represent the true class in the image.

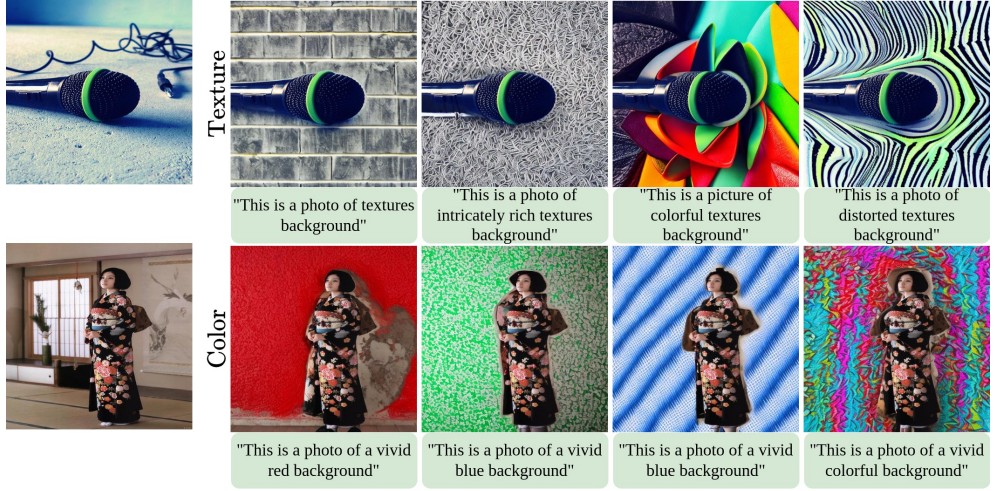

Figure 15: The figure illustrates the introduction of background variations achieved through a diverse set of texture and color text prompts

## A.5 QUALITATIVE RESULTS ON DETECTION

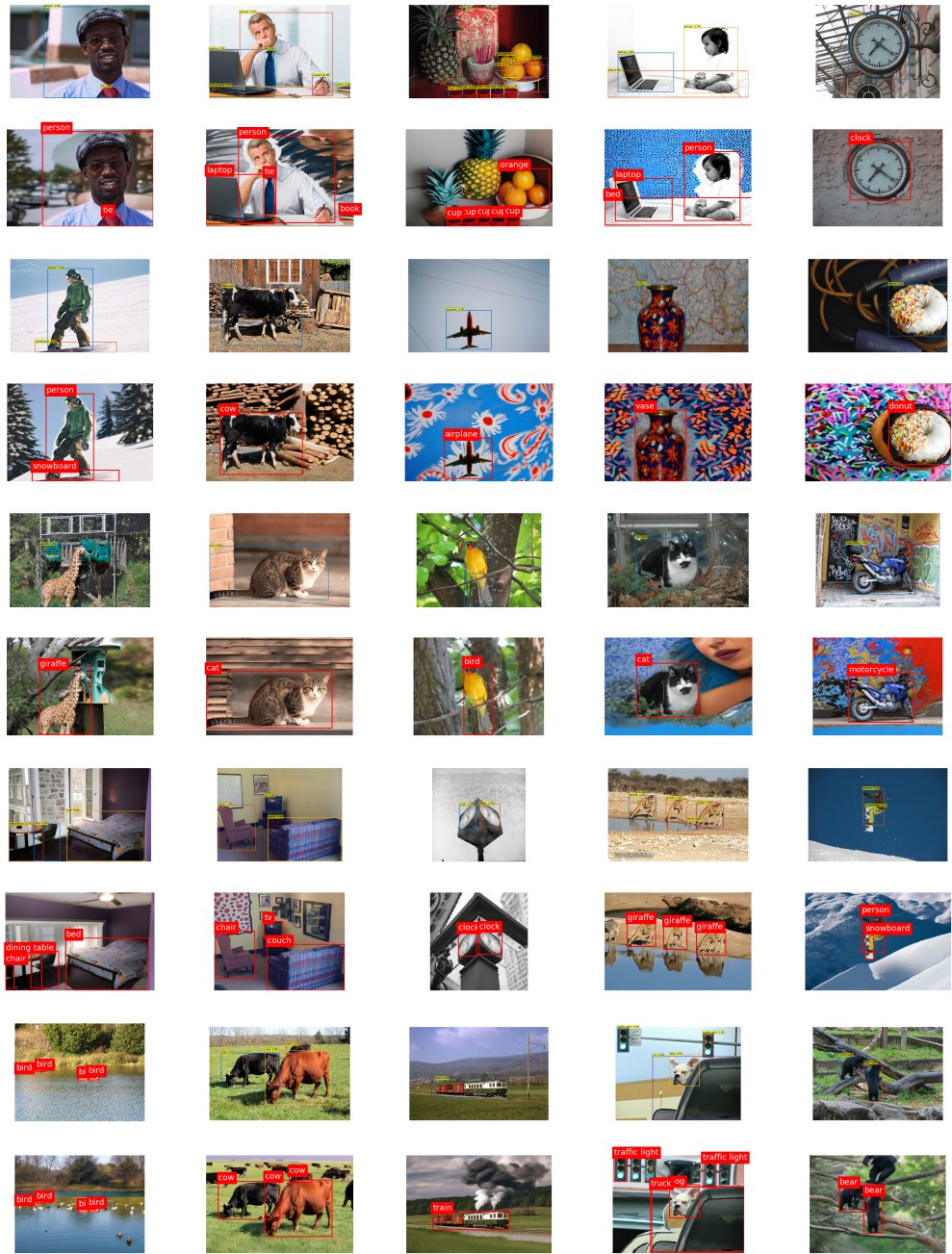

Figure 16: We use diverse prompts to capture the diverse background shifts on samples from COCO-DC. The figure illustrate a comparison of prediction of Mask-RCNN on both clean and generated samples on COCO-DC. Each two adjacent rows represents the prediction of Mask-RCNN on clean *(top)* and generated images *(bottom)*.

Table 8: IoU distribution of FastSAM. We report the percentage of images that have IoU within the given range.

| Background | 0.0-0.2 | 0.2-0.4 | 0.4-0.6 | 0.6-0.8 | 0.8-1.0 |
|---|---|---|---|---|---|
| Class Label | 8.10 | 5.93 | 8.02 | 13.03 | 64.92 |
| BLIP-2 Caption | 5.70 | 4.81 | 6.92 | 13.01 | 69.56 |
| Color | 1.65 | 1.39 | 2.31 | 4.99 | 89.65 |
| Texture | 2.11 | 1.02 | 1.78 | 4.07 | 91.02 |
| Adversarial | 4.87 | 2.91 | 4.32 | 10.63 | 77.27 |

Table 9: DETR Object detection evaluation on `COCO-DC`

| Background | Box AP | Recall AR |
|---|---|---|
| Original | 0.65 | 0.81 |
| BLIP-2 Caption | 0.53 | 0.76 |
| Color | 0.52 | 0.73 |
| Texture | 0.52 | 0.71 |
| Adversarial | 0.42 | 0.62 |

## A.6 EFFECT OF BACKGROUND CHANGE ON SEGMENTATION MODELS

Figure 17, 18, and 19 provide failure cases of FastSAM to correctly segment the object in the images where background has been changed in terms of color, texture, and adversarial, respectively. Since we obtain the object masks for `IN-Nat` using FastSAM, we compare those masks using IoU with the ones obtained by FastSAM on the generated dataset (see Table 8).

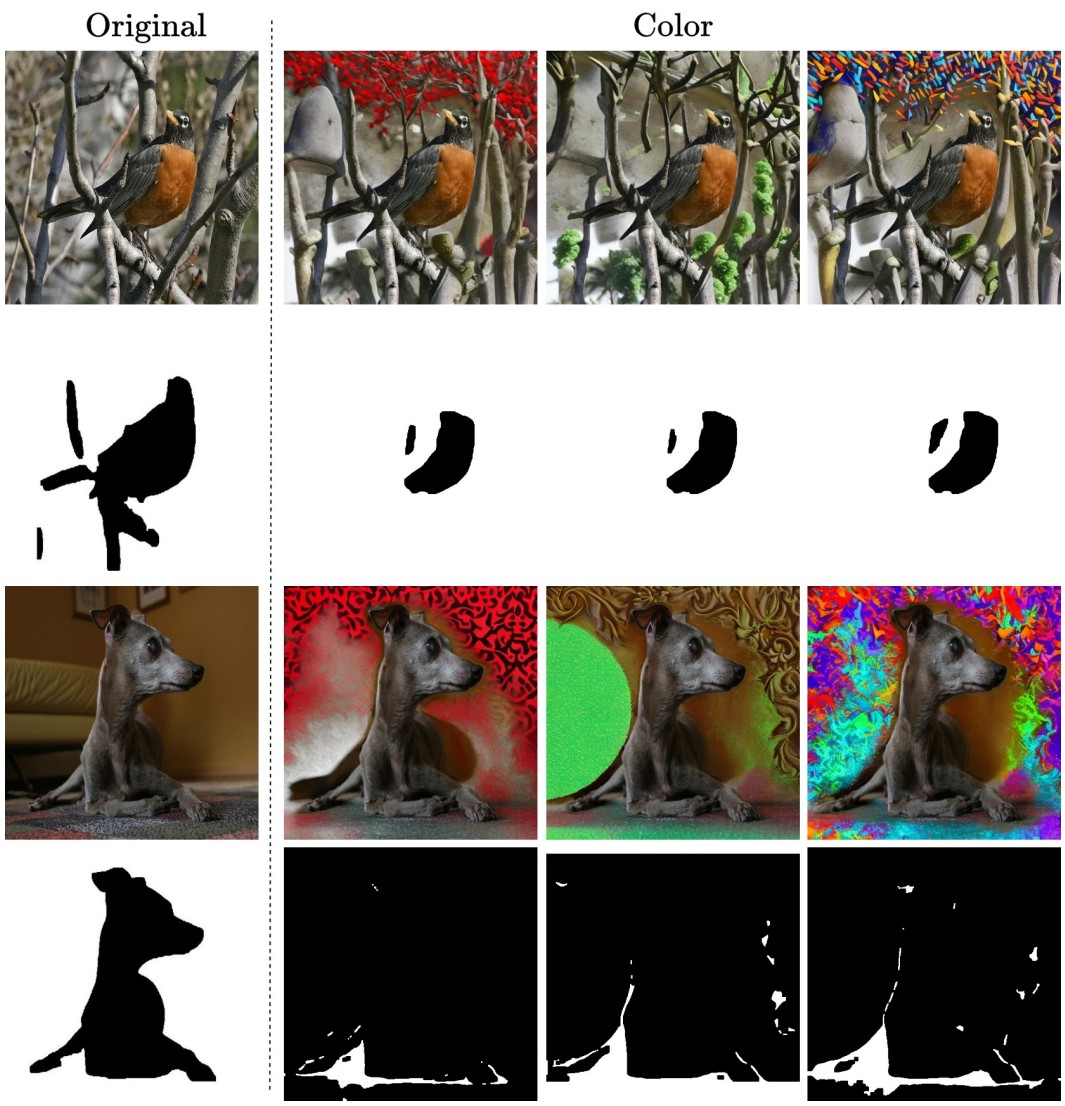

Figure 17: Instances illustrating FastSAM model's failure to accurately segment masks for the background color changes on `IN-Nat` samples..

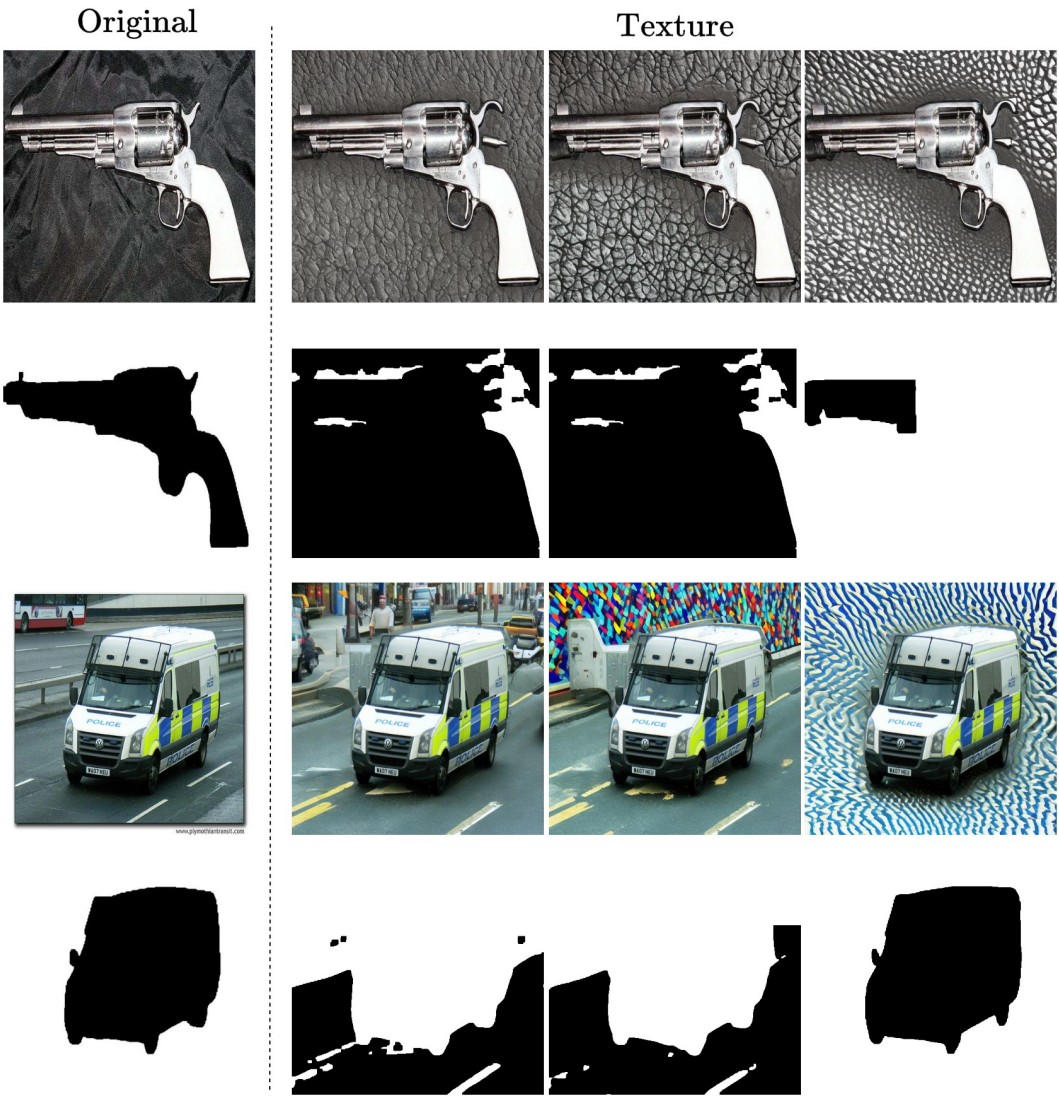

Figure 18: Instances illustrating FastSAM model's failure to accurately segment masks for the background texture changes on `IN-Nat` samples.

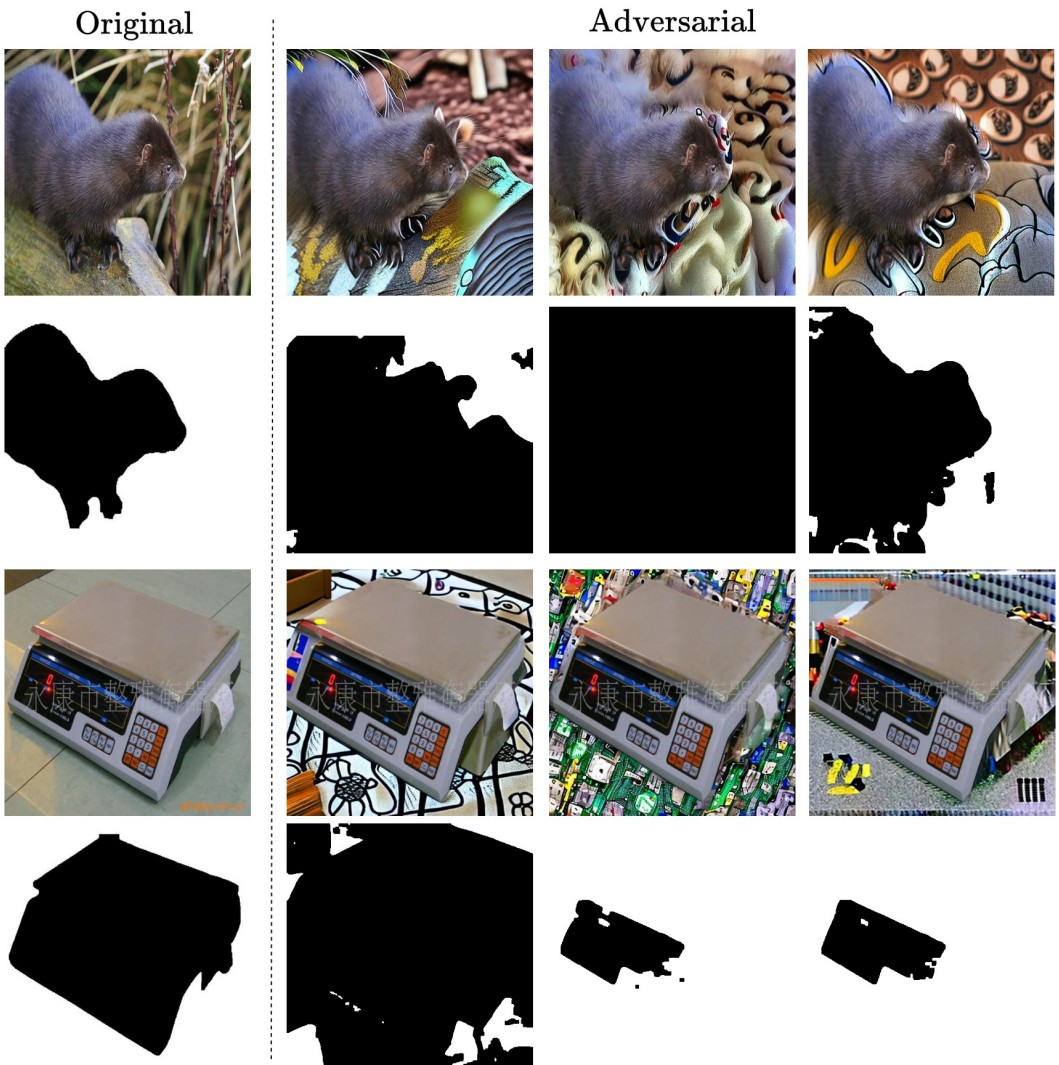

Figure 19: Instances illustrating FastSAM model's failure to accurately segment masks for the adversarial background changes on `IN-Adv` samples.

Table 10: Performance evaluation of naturally trained classifiers and zero-shot CLIP models on `IN-Nat`. The text prompts used for color and texture changes are provided in Table 6.

| Background | Naturally Trained Models | | | | | | | |
| --- | --- | --- | --- | --- | --- | --- | --- | --- |
| | ViT | | | | CNN | | | |
| | ViT-T | ViT-S | Swin-T | Swin-S | ResNet50 | ResNet152 | DenseNet | Average |
| Clean | 96.04 | 98.18 | 98.65 | 98.84 | 98.65 | 99.27 | 98.09 | 98.25 |
| Color$_{prompt-1}$ | 76.58 | 86.43 | 88.92 | 91.23 | 91.08 | 93.79 | 86.05 | 87.72 |
| Color$_{prompt-2}$ | 77.09 | 87.57 | 89.33 | 90.99 | 90.62 | 93.40 | 86.61 | 87.94 |
| Color$_{prompt-3}$ | 76.80 | 86.97 | 88.74 | 90.99 | 90.62 | 93.18 | 87.41 | 87.82 |
| Color$_{prompt-4}$ | 70.64 | 84.52 | 86.84 | 88.84 | 89.44 | 92.89 | 83.19 | 85.19 |
| Texture$_{prompt-1}$ | 79.07 | 87.92 | 90.17 | 91.68 | 91.18 | 94.42 | 88.28 | 88.96 |
| Texture$_{prompt-2}$ | 75.29 | 85.84 | 87.74 | 90.32 | 89.01 | 93.04 | 84.77 | 86.57 |
| Texture$_{prompt-3}$ | 67.97 | 82.54 | 86.17 | 87.99 | 87.99 | 91.28 | 82.99 | 83.85 |
| Texture$_{prompt-4}$ | 68.24 | 79.73 | 81.09 | 84.41 | 83.21 | 87.66 | 77.29 | 80.23 |

| Background | CLIP Models | | | | | | | |
| --- | --- | --- | --- | --- | --- | --- | --- | --- |
| | ViT | | | CNN | | | | |
| | ViT-B/32 | ViT-B/16 | ViT-L/14 | ResNet50 | ResNet101 | ResNet50x4 | ResNet50x16 | Average |
| Clean | 75.56 | 81.56 | 88.61 | 73.06 | 73.95 | 77.87 | 83.25 | 79.12 |
| Color$_{prompt-1}$ | 58.32 | 65.54 | 72.75 | 57.43 | 60.92 | 65.97 | 73.04 | 64.49 |
| Color$_{prompt-2}$ | 57.91 | 67.28 | 74.44 | 58.67 | 60.12 | 65.9 | 74.13 | 65.49 |
| Color$_{prompt-3}$ | 57.27 | 66.77 | 74.07 | 57.89 | 59.03 | 66.10 | 74.06 | 65.03 |
| Color$_{prompt-4}$ | 53.02 | 63.08 | 71.42 | 53.53 | 55.87 | 60.05 | 71.28 | 61.18 |
| Texture$_{prompt-1}$ | 59.05 | 68.50 | 75.67 | 60.38 | 61.78 | 66.99 | 74.31 | 66.67 |
| Texture$_{prompt-2}$ | 58.60 | 68.01 | 74.40 | 58.29 | 59.56 | 66.34 | 74.67 | 65.69 |
| Texture$_{prompt-3}$ | 52.89 | 64.30 | 68.70 | 53.29 | 55.35 | 61.58 | 69.35 | 60.78 |
| Texture$_{prompt-4}$ | 51.01 | 62.25 | 69.08 | 51.35 | 53.46 | 61.10 | 70.33 | 59.79 |

Table 11: Performance evaluation of naturally trained classifiers on `COCO-DC` dataset. The text prompts used for color and texture changes are provided in Table 6.

| Background | ViT | | | | CNN | | |
| --- | --- | --- | --- | --- | --- | --- | --- |
| | ViT-T | ViT-S | Swin-T | Swin-S | ResNet50 | DenseNet-161 | Average |
| Clean | 82.96 | 86.24 | 88.55 | 90.23 | 88.55 | 86.77 | 87.21 |
| Color$_{prompt-1}$ | 61.66 | 65.92 | 73.38 | 73.73 | 75.86 | 71.6 | 70.35 |
| Color$_{prompt-2}$ | 64.86 | 70.01 | 76.84 | 77.10 | 77.81 | 75.06 | 73.61 |
| Color$_{prompt-3}$ | 62.64 | 67.52 | 73.29 | 74.09 | 77.28 | 73.64 | 71.41 |
| Color$_{prompt-4}$ | 55.54 | 61.04 | 70.09 | 72.13 | 74.97 | 66.19 | 66.66 |
| Texture$_{prompt-1}$ | 67.96 | 70.36 | 75.42 | 78.70 | 79.94 | 73.55 | 74.32 |
| Texture$_{prompt-2}$ | 63.97 | 69.56 | 74.62 | 77.72 | 78.97 | 75.15 | 73.33 |
| Texture$_{prompt-3}$ | 52.52 | 58.82 | 68.05 | 70.09 | 70.71 | 63.79 | 63.99 |
| Texture$_{prompt-4}$ | 56.16 | 61.57 | 66.72 | 70.18 | 69.56 | 67.25 | 65.24 |

## A.7 EVALUATION ON COLOR AND TEXTURE PROMPTS

In this section, we provide evaluation across diverse color and texture prompts. In Table 10 uni-modal classifier and multi-modal CLIP models are evaluated on `IN-Nat`. Furthermore, in Table 11 we extend the analysis to report results on `COCO-DC` for the classification task. For ease of comparison we plot the results obtained on uni-modal models and multi-models on `IN-Nat` dataset side-by-side in Figure 20.

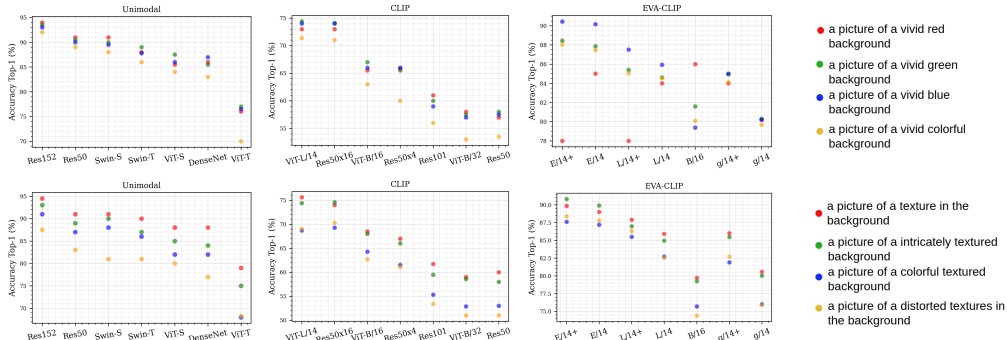

Figure 20: We compare the performance of uni-modal, multi-modal CLIP and EVA-CLIP on diverse color and texture prompts. We observe across Uni-modal models, CNNs and hybrid models tend to perform better than pure transformer based models with similar capacity. EVA-CLIP perform significantly better than the CLIP models even for models with similar capacity (ViT-L). The plots are evaluated on `IN-Nat` dataset under texture and color changes.

## A.8 EXPLORING FEATURE SPACE OF VISION MODELS

In Figure 21 and 22, we explore the visual feature space of vision and vision language model using t-SNE visualizations. We observe that as the background changes deviate further from the original background, a noticeable shift occurs in the feature space. The distinct separation or clustering of features belonging to the same class appears to decrease. This observation suggests a significant correlation between the model's decision-making process and the alterations in the background. Furthermore, we also show the GradCAM (Selvaraju et al., 2019) on generated background changes. We observe that diverse background changes significantly shift the attention of the model as can be seen from Figure 23 and 24.

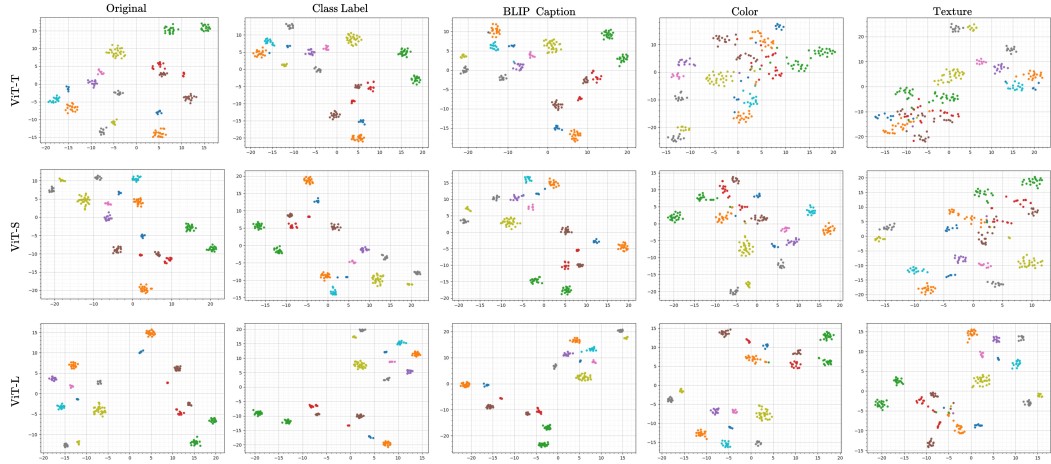

Figure 21: t-SNE visualization of classifier models on `IN-Nat` dataset.

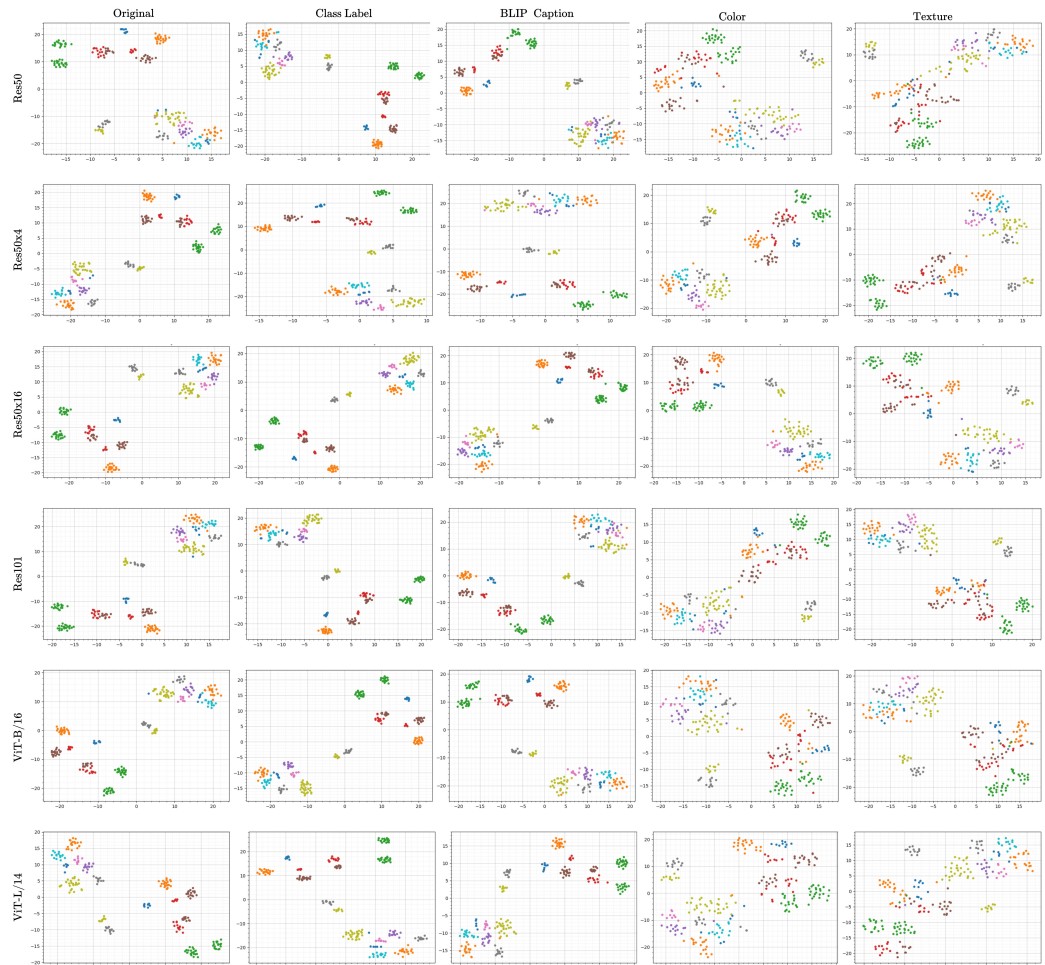

Figure 22: t-SNE visualization of CLIP Vision Encoder features on `IN-Nat` dataset.

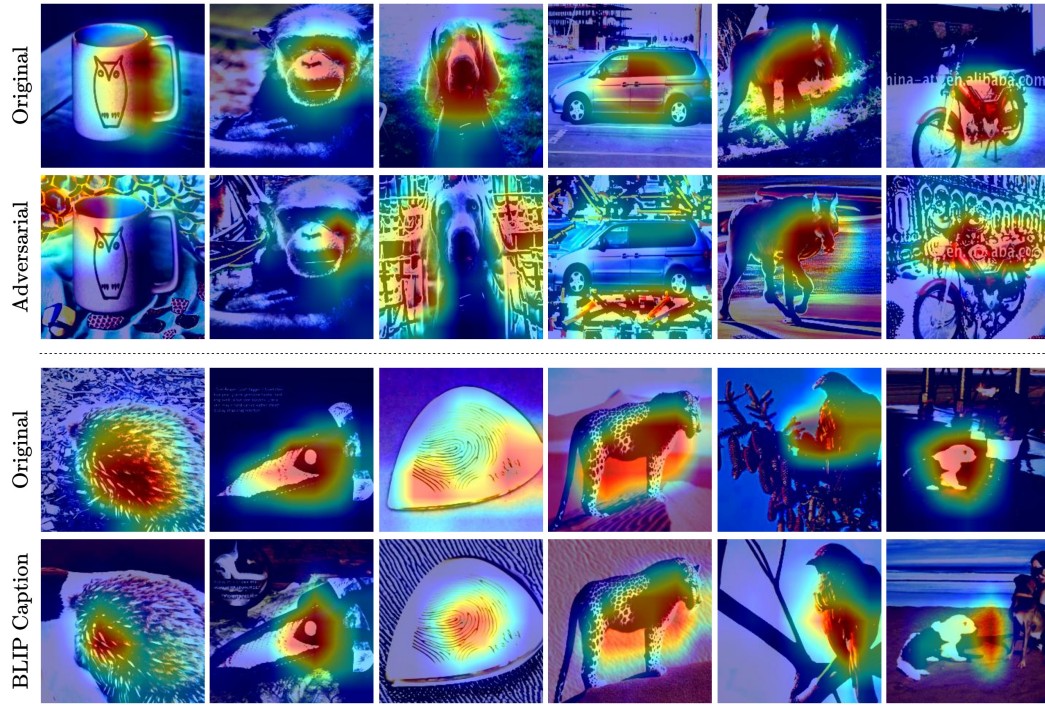

Figure 23: GradCAM (Selvaraju et al., 2019) visualization of adversarial and BLIP-2 background examples. The activation maps were generated on ImageNet pre-trained Res-50 model.

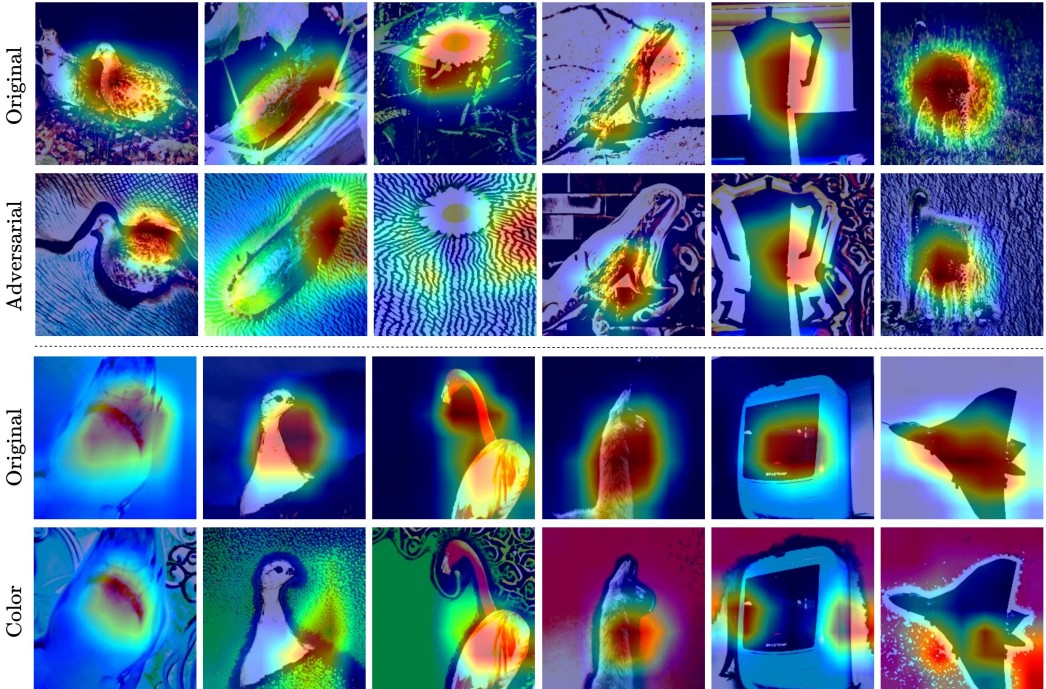

Figure 24: GradCAM (Selvaraju et al., 2019) visualization of texture and color background changes. The activation maps were generated on ImageNet pre-trained Res-50 model.

Table 12: Performance evaluation and comparison of our dataset with LANCE counterfactual images. The images are generated on `IN-Adv` dataset.

| Background | ViT | | | | CNN | | | Average |
|---|---|---|---|---|---|---|---|---|
| | ViT-T | ViT-S | Swin-T | Swin-S | Res-50 | Res-152 | DenseNet-161 | |
| Original | 95.5 | 97.5 | 97.9 | 98.3 | 98.5 | 99.1 | 97.2 | 97.71 |
| Class label | 90.5 | 94.0 | 95.1 | 95.4 | 96.7 | 96.5 | 94.7 | 94.70 |
| BLIP-2 Caption | 85.5 | 89.1 | 91.9 | 92.1 | 93.9 | 94.5 | 90.6 | 91.08 |
| Color | 67.1 | 83.8 | 85.8 | 86.1 | 88.2 | 91.7 | 80.9 | 83.37 |
| Texture | 64.7 | 80.4 | 84.1 | 85.8 | 85.5 | 90.1 | 80.3 | 81.55 |
| Adversarial | 18.4 | 32.1 | 25.0 | 31.7 | 2.0 | 14.0 | 28.0 | 21.65 |
| LANCE (Prabhu et al., 2023) | 80.0 | 83.8 | 87.6 | 87.7 | 86.1 | 87.4 | 85.1 | 85.38 |

## A.9 COMPARISON WITH RELATED WORKS

Prabhu et al. (2023) proposed the LANCE method, which is closely relevant to our approach. LANCE leverages the capabilities of language models to create textual prompts, facilitating diverse image alterations using the prompt-to-prompt image editing method (Hertz et al., 2022a) and null-text inversion (Mokady et al., 2023) for real image editing. However, this reliance on prompt-to-prompt editing imposes constraints, particularly limiting its ability to modify only specific words in the prompt. Such a limitation restricts the range of possible image transformations. Additionally, the global nature of their editing process poses challenges in preserving object semantics during these transformations. In contrast, our method employs both visual and textual conditioning, effectively preserving object semantics while generating varied background changes. This approach aligns well with our goal of studying object-to-background context. We use open-sourced code from LANCE to compare it against our approach both quantitatively and qualitatively. We use a subset of 1000 images, named `IN-Adv`, for comparison. **a) Quantitative comparison:** We observe that our natural object-to-background changes including color and texture perform favorably against LANCE, while our adversarial object-to-background changes perform significantly better as shown in the Table 12. **b) Qualitative comparison:** Since LANCE relies on global-level image editing, thus tends to alter the object semantics and distort the original object shape in contrast to our approach which naturally preserves the original object and alters the object-to-background composition only. This can be observed in qualitative examples provided in Figures 25 and 26. We further validate this effect by masking the original images and LANCE-generated counterfactual images. As reported in Table 13, we observe an accuracy drop from $97.71\%$ to $84.35\%$ across different models. Note that this is the first step of our approach, where the background is just masked but not optimized for any background changes. However, when the background is masked in LANCE-generated counterfactual images, overall accuracy drops from $97.71\%$ to $71.57\%$. This shows that the LANCE framework has changed the original object during optimization which is also reflected in Figures 25 and 26. Therefore, our proposed approach allows us to study the correlation of object to background changes without distorting the original object.

Table 13: Performance evaluation and comparison on `IN-Adv` dataset. The drop in accuracy of LANCE dataset when the background is masked clearly highlight the image manipulation being done on the object of interest.

| Masked Background | Foreground | | | | | | | Average |
|---|---|---|---|---|---|---|---|---|
| | ViT-T | ViT-S | Swin-T | Swin-S | Res-50 | Res-152 | DenseNet-161 | |
| Original | 70.5 | 86.1 | 84.2 | 87.6 | 87.2 | 91.2 | 83.7 | 84.35 |
| LANCE (Prabhu et al., 2023) | 59.5 | 72.5 | 72.3 | 75.3 | 71.9 | 77.5 | 72.0 | 71.57 |

Original

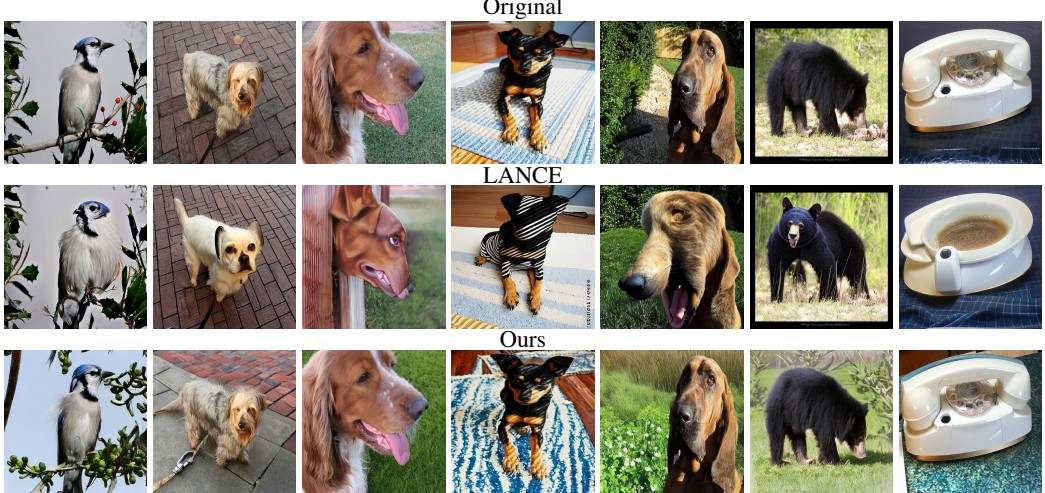

LANCE

Ours

Figure 25: Visual illustrations of generated images on `IN-ADV` dataset using LANCE's prompts for background variation. LANCE fails to preserve object semantics, limiting our ability to evaluate models on the background alone. In contrast, our method exclusively edits the background.

Original

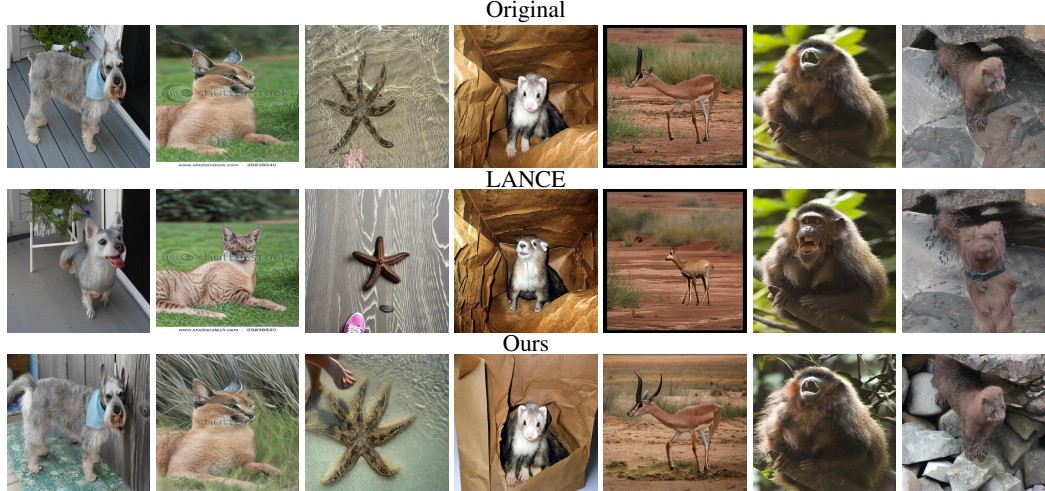

LANCE

Ours

Figure 26: Visual illustrations of generated images on `IN-ADV` dataset using LANCE's prompts for background variation. LANCE fails to preserve object semantics, limiting our ability to evaluate models on the background alone. In contrast, our method exclusively edits the background.

### A.10 MISCLASSIFIED SAMPLES

We observe that there exist images which get misclassified *(by ResNet-50)* across several background alterations as can be seen from from Figure 27. In Figure 28 we show examples on which the highly robust *EVA-CLIP* ViT-E/14+ model fails to classify the correct class. After going through the misclassified samples, we visualize some of the *hard* examples in Figure 29. Furthermore, we also provide visualisation of images misclassified with adversarial background changes in Figure 30.

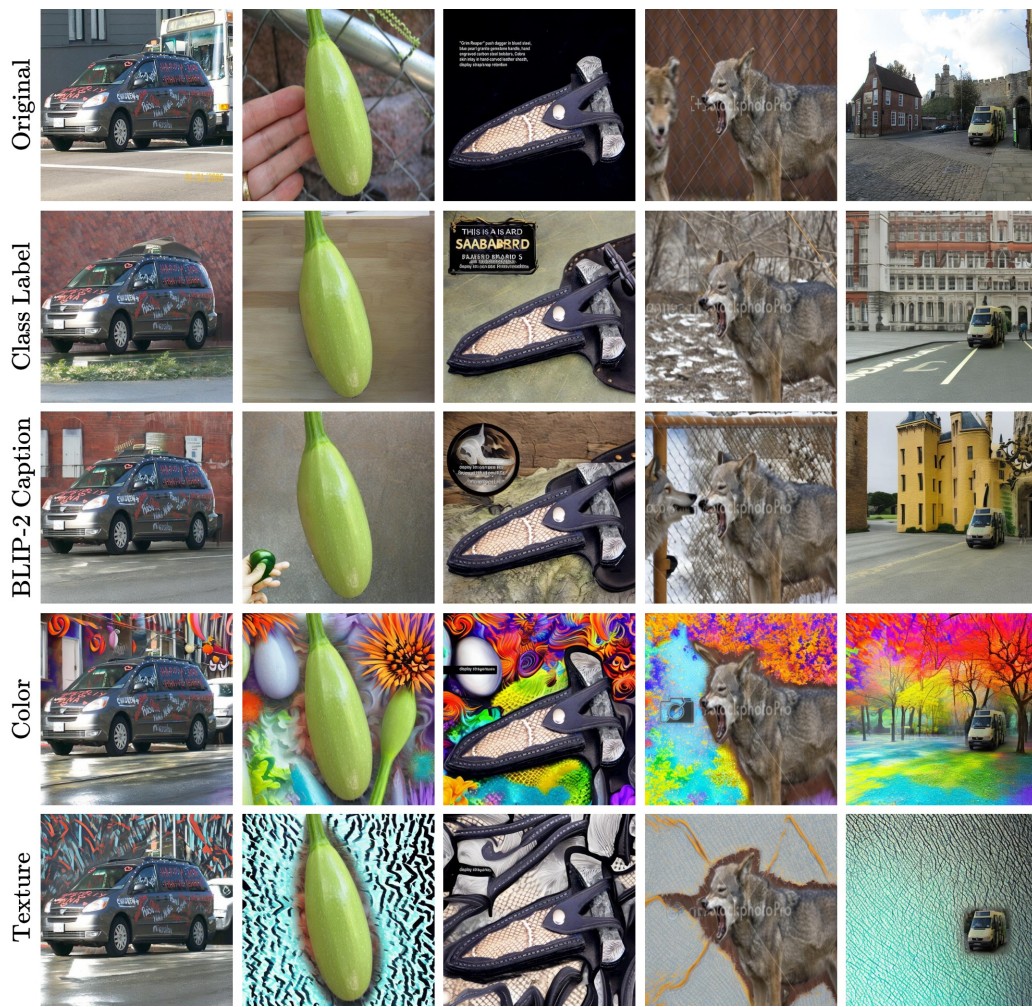

Figure 27: Images misclassified by Res-50 across different background changes

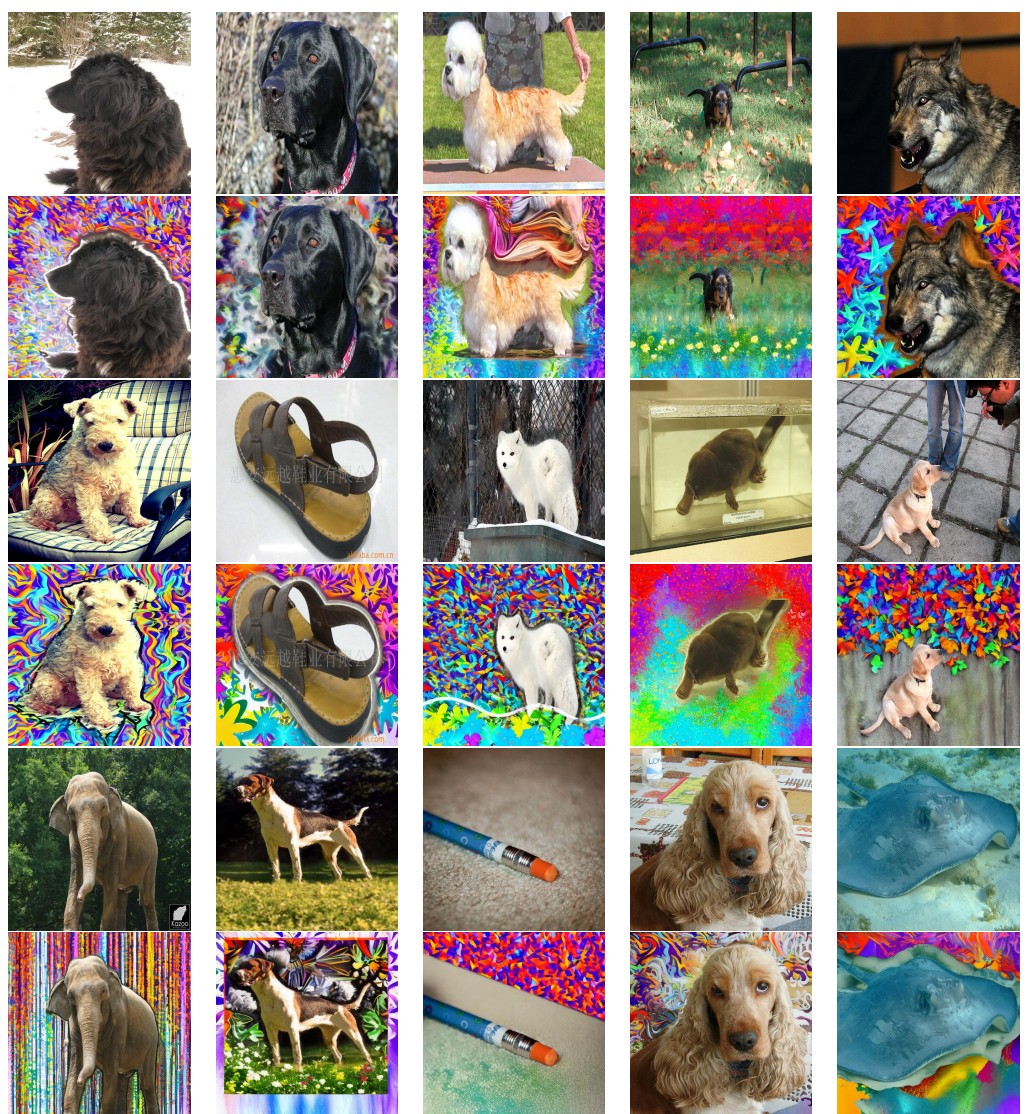

Figure 28: Visual illustration of misclassified samples on color background and corresponding clean image samples. In two adjacent rows, *first row* represent the clean images and the *second row* represent the corresponding colorful background images

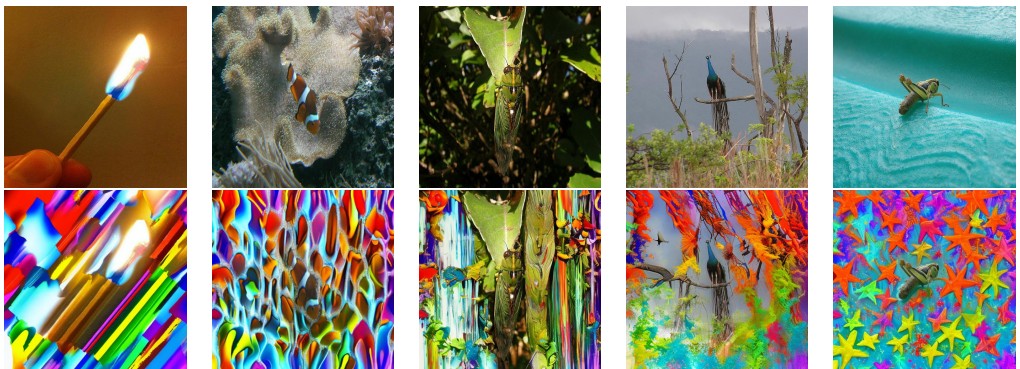

Figure 29: Visual illustration of *hard* samples on color background

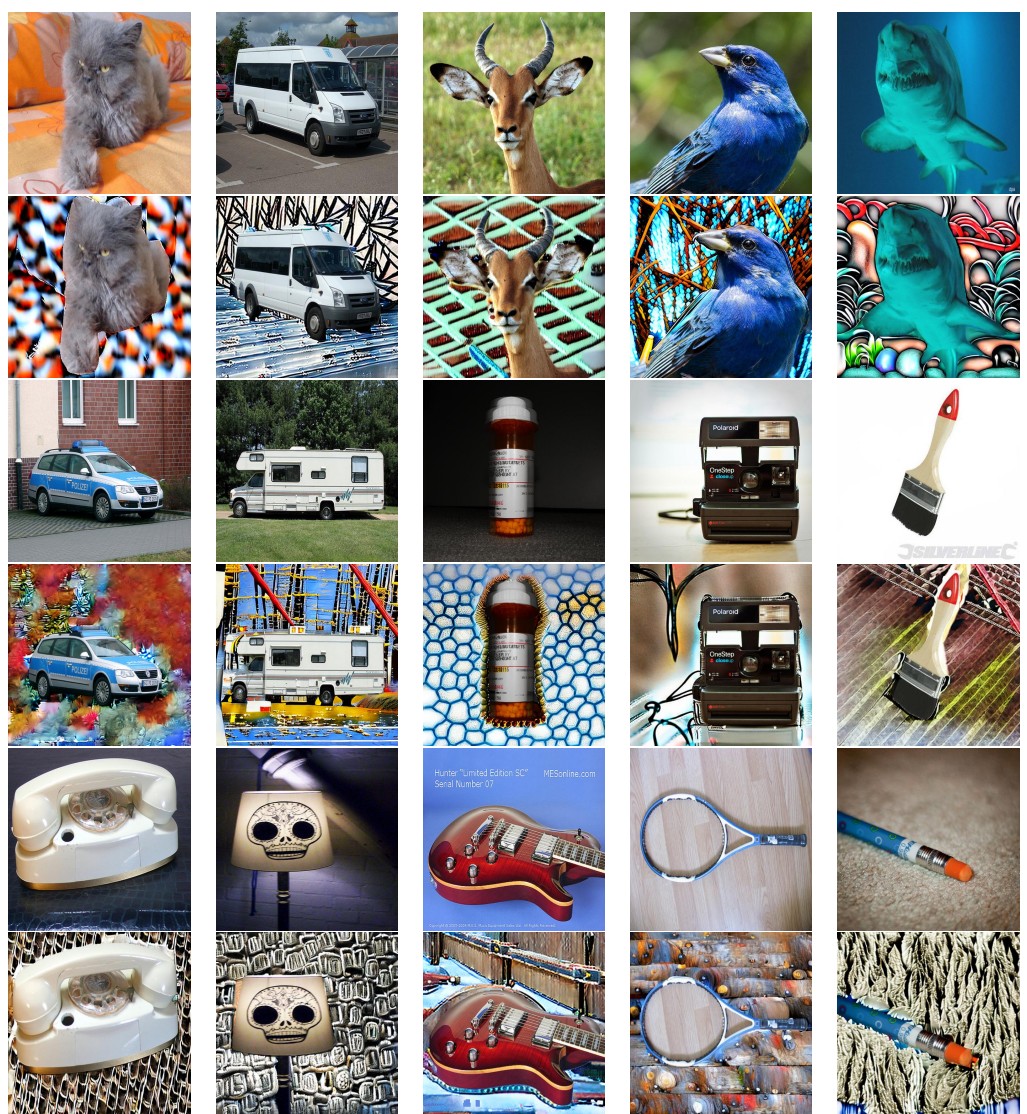

Figure 30: Visual illustration of misclassified samples on adversarial background and corresponding clean image samples. In two adjacent rows, *first row* represents the clean images and the *second row* represents the corresponding adversarial images

### A.11 LIMITATIONS AND FUTURE DIRECTIONS

**Limitations:** In Figure 31, we observe that for objects covering a small region in the image, relying solely on the class name to guide the diffusion model can result in alterations of the object shape, expanding the influence of the class name semantics to larger image regions. However, by supplementing with descriptive captions that encompass the object-to-background context, we partially mitigate this effect. Furthermore, the generated textured background can inadvertently camouflage the object. To address this concern, we slightly expand the object mask to clearly delineate the object boundaries.

**Future Directions:** Our current work represents one of the preliminary efforts in utilizing diffusion models to study the object-to-background context in vision-based systems. Based on our observations and analysis, the following are the interesting future directions.

- Since large capacity models in general show better robustness to object-to-background compositions, coming up with new approaches to effectively distill knowledge from these large models could improve how small models cope with background changes. This can improve resilience in small models that can be deployed in edge devices.

- Another direction is to set up object-to-background priors during adversarial training to expand robustness beyond just adversarial changes. To some extent, successful examples are recent works (Sitawarin et al., 2022; Darcet et al., 2023) where models are trained to discern the salient features in the image foreground. This leads to better robustness.

- Our work can be extended to videos where preserving the semantics of the objects across the frames while introducing changes to the background temporally will help understand the robustness of video models.

- Additionally, the capabilities of diffusion models can be explored to craft complex changes in the object of interest while preserving the semantic integrity. For instance, in Yuan et al. (2023), diffusion models are employed to generate multiple viewpoints of the same object. Additionally, in (Kawar et al., 2023), non-rigid motions of objects are created while preserving their semantics. By incorporating these with our approach, we can study how vision models maintain semantic consistency in dynamic scenarios.

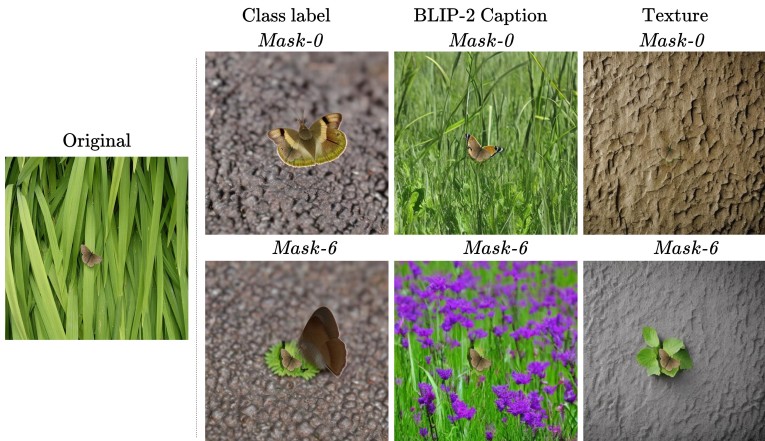

Figure 31: Limitation: Background changes on small objects in the scene. Enlarging the mask (here by 6 pixels) helps in mitigating the issue to some effect.

### A.12 DATASET DISTRIBUTION

`IN-Nat` dataset comprises a wide variety of objects belonging to different classes, as illustrated in Figure 32. Our dataset maintains a clear distinction between the background and objects, achieved through a rigorous filtering process applied to the ImageNet validation dataset. Additionally, we provide the list of prompts in Table 6 utilized for the experiments.

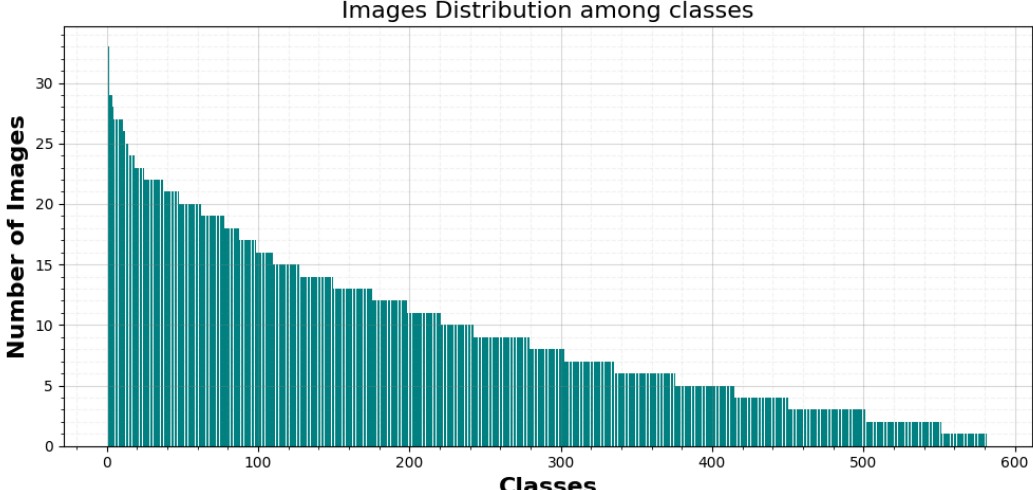

Figure 32: Our `IN-Nat` dataset encompasses a diverse variety of images spanning 582 distinct classes. In this illustration, we showcase images distribution among all the classes. The figure is plotted in decreasing order of images present in each class.

## A.13 EVALUATION ON RECENT VISION MODELS

We have conducted experiments on recent transformer CNN based models like DeiT (Touvron et al., 2021) and ConvNeXt (Liu et al., 2022), and their results are presented in Table 14. We observe a consistent trend in model performance on our dataset, revealing that even the modern vision models are vulnerable to background changes.

Table 14: Performance evaluation on naturally trained classifiers on `IN-NAT` and `IN-ADV` dataset. All models exhibit a marked decrease in accuracy when the background is modified, highlighting their sensitivity to changes in the environment. The decline in performance is minimal with class label backgrounds but more pronounced with texture and color alterations. The most significant accuracy drop occurs under adversarial conditions, underscoring the substantial challenge posed by such backgrounds to the classifiers.

| Datasets | Background | Transformers | | | | CNN | | | |
|---|---|---|---|---|---|---|---|---|---|
| | | DeiT-T | DeiT-S | DeiT-B | Average | ConvNeXt-T | ConvNeXt-B | ConvNeXt-L | Average |
| IN-Nat | Original | 96.36 | 99.27 | 99.41 | 98.34 | 99.07 | 99.21 | 99.40 | 99.22 |
| | Class label | 94.18 | 96.85 | 97.74 | 96.25 | 97.60 | 97.51 | 97.51 | 97.54 |
| | BLIP-2 Caption | 89.33 | 94.29 | 95.07 | 92.89 | 94.64 | 94.82 | 95.47 | 94.97 |
| | Color | 80.96 | 89.48 | 91.11 | 87.13 | 92.11 | 93.58 | 93.58 | 93.09 |
| | Texture | 74.15 | 84.01 | 86.75 | 81.63 | 88.50 | 89.50 | 91.13 | 89.71 |
| IN-ADV | Original | 95.44 | 99.10 | 99.10 | 97.88 | 99.00 | 99.00 | 92.92 | 96.97 |
| | Adversarial | 20.40 | 29.62 | 34.81 | 28.27 | 32.88 | 42.52 | 48.60 | 41.33 |

## A.14 EVALUATION ON DINOv2 MODELS

Our findings underscore the necessity of training vision models to prioritize discriminative and salient features, thereby diminishing their dependence on background cues. Recent advancements, such as the approaches by Sitawarin et al. (2022) employing a segmentation backbone for classification to improve adversarial robustness and by Darcet et al. (2023) using additional learnable tokens known as *registers* for interpretable attention maps, resonate with this perspective. Our preliminary experiments with the DINOv2 models Oquab et al. (2023), as presented in Table 15, corroborate this hypothesis. Across all the experiments, models with registers *(learnable tokens)* provide more robustness to background changes, with significant improvement seen in the adversarial background changes.

Table 15: Performance comparison of classifiers that are trained different on `IN-NAT` dataset. The DINOv2 model with registers generally shows higher robustness to background changes, particularly in the presence of color, texture and adversarial backgrounds. This suggests that the incorporation of registers in DINOv2 enhances its ability to maintain performance despite challenging background alterations.

| Dataset | Background | Dinov2 | | | | Dinov2$_{registers}$ | | | |
|---|---|---|---|---|---|---|---|---|---|
| | | ViT-S | ViT-B | ViT-L | Average | ViT-S | ViT-B | ViT-L | Average |
| IN-Nat | Original | 96.78 | 97.18 | 98.58 | 97.51 | 97.71 | 98.05 | 99.14 | 98.30 |
| | Class label | 94.62 | 96.02 | 97.18 | 95.94 | 95.55 | 96.44 | 97.94 | 96.64 |
| | BLIP-2 Caption | 89.22 | 91.73 | 94.33 | 91.76 | 90.86 | 92.10 | 95.02 | 92.66 |
| | Color | 83.85 | 89.68 | 93.31 | 88.94 | 85.88 | 91.15 | 94.64 | 90.55 |
| | Texture | 83.63 | 89.08 | 92.44 | 88.38 | 84.98 | 91.03 | 93.97 | 89.99 |
| IN-Adv | Original | 95.12 | 96.50 | 98.10 | 96.57 | 97.91 | 97.80 | 99.00 | 98.23 |
| | Adversarial | 54.31 | 71.62 | 80.87 | 68.93 | 58.30 | 76.21 | 84.50 | 73.00 |

## A.15 EVALUATION ON BACKGROUND/FOREGROUND IMAGES

In this section, we systematically evaluate vision-based models by focusing on background and foreground elements in images. This evaluation involves masking the background of the original image, allowing us to assess the model's performance in recognizing and classifying the foreground without any cues from the background context. Conversely, we also mask the object or foreground from the image. This step is crucial to understand to what extent the models rely on background information for classifying the image into a specific class. This dual approach provides a comprehensive insight into the model's capabilities in image classification, highlighting its reliance on foreground and background elements.

Table 16: Evaluation of Zero-shot CLIP Models on `IN-NAT` dataset while masking the object or the background of the image. Top-1(%) accuracy is reported. The accuracy drop is observe when we remove the object clues from the background such as in texture or color background

| Background | Foreground | | | | | | | |
|---|---|---|---|---|---|---|---|---|
| | Res50 | Res101 | Res50x4 | Res50x16 | ViT-B/32 | ViT-B/16 | ViT-B/14 | Average |
| Original | 54.76 | 58.89 | 64.86 | 70.80 | 59.47 | 69.42 | 79.12 | 65.33 |
| | **Background** | | | | | | | |
| Original | 15.84 | 17.74 | 18.47 | 20.67 | 17.72 | 21.28 | 28.99 | 20.10 |
| Class label | 27.17 | 29.08 | 33.02 | 35.93 | 31.35 | 38.74 | 46.88 | 34.59 |
| BLIP-2 Caption | 19.05 | 21.39 | 23.37 | 24.57 | 22.39 | 27.21 | 34.42 | 24.62 |
| Color | 3.92 | 5.46 | 5.64 | 6.53 | 5.64 | 6.95 | 10.28 | 6.34 |
| Texture | 3.65 | 5.12 | 5.12 | 5.84 | 5.43 | 6.68 | 10.04 | 5.98 |

Table 17: DINOv2 model evaluation by masking either the object or the background within the `IN-NAT` dataset. The integration of the additional token in the DINOv2 model proves beneficial, contributing to enhanced accuracy. However, our observations reveal that these models remain susceptible to background cues, particularly evident in class labels and the BLIP-2 Caption dataset. Interestingly, as we transition towards more generic texture or color backgrounds, a discernible drop in accuracy is observed.

| Background | Foreground | | | | | | | |
|---|---|---|---|---|---|---|---|---|
| | ViT-S | ViT-B | ViT-L | Average | ViT-S$_{reg}$ | ViT-B$_{reg}$ | ViT-L$_{reg}$ | Average |
| Original | 88.73 | 93.86 | 94.89 | 92.49 | 96.34 | 89.95 | 97.25 | 94.51 |
| | **Background** | | | | | | | |
| Original | 27.72 | 37.78 | 51.44 | 38.98 | 30.10 | 42.08 | 55.18 | 42.45 |
| Class label | 42.70 | 54.73 | 66.68 | 54.70 | 46.88 | 58.81 | 68.97 | 58.22 |
| BLIP-2 Caption | 30.51 | 40.74 | 50.57 | 40.60 | 33.62 | 42.48 | 52.40 | 42.83 |
| Color | 2.96 | 5.03 | 8.39 | 5.46 | 3.68 | 5.75 | 9.50 | 6.31 |
| Texture | 2.83 | 4.92 | 7.88 | 5.21 | 3.45 | 5.57 | 9.28 | 6.10 |