# OpenReview forum: "BackBench: Are Vision Language Models Resilient to Object-to-Background Context?"
_ICLR.cc/2024/Conference — ICLR 2024 Conference Withdrawn Submission_

### Official Review · Reviewer_SPdM · 2023-10-29

**Soundness:** 2 fair
**Presentation:** 2 fair
**Contribution:** 2 fair
**Rating:** 5
**Confidence:** 4

**Summary:**

The goal of this paper is to evaluate the resilience of modern vision and multi-modal foundation models against object-to-background context variations. The authors developed a pipeline that can change the background of real images while preserving the foreground semantics. The generated data allows a in-depth analysis of the robustness of vision models.

**Strengths:**

1. This work focuses on an important topic about the robustness of vision and vision-language models to object-to-background context. Such benchmarks allow us to analyze the limitations of existing models.
2. The method is built on pretrained models, such as SAM, BLIP2 and Stable Diffusion. Making use of these pretrained models for synthetic data generation/perturbation are interesting.

**Weaknesses:**

1. The generation pipeline is built on existing methods. The whole method is not entirely new. It seems that the only changes being made are appending some pre-defined descriptions of the background. Figure 3 clearly presents a more diverse backgrounds than other examples in the supplementary results. I believe there’s more to explore how to generate diverse backgrounds other than randomizing seeds.
2. The title discusses the robustness of vision language models while most experiments focus on classification and detection, and there’s only one real multi-modal task, image captioning.
3. Latest transformer/CNN models should be considered, such as DeiT and ConvNeXt.
4. There are limited analyses of the experimental results. A lot of results are presented but what are the conclusions?
5. References to SAM and BLIP-2 should be added in Section 3.2.
6. The presentation of the paper can be improved. Some technical details are not very clear in Section 3.2 (see Question 1) and it’s unclear what changes have been made other than directly applying the existing models (see Question 2). Also the presentation of Section 4 is a bit messy to me. A lot of experimental settings and results are presented while there is only one sub-section “4.1 Results”.

**Questions:**

1. What “changes” are made from $\mathcal{T}$ to $\mathcal{T}’$? Just appending prompts about the background in Table 6?
2. In Section 3.2, are there any changes been made besides applying the inpainting diffusion model?

---

> ### Author Response · Authors · 2023-11-22
> **Response to Reviewer SPdM (1/3)**
>
> We thank the reviewer for the insightful comments. Please find our responses to specific queries below.
>
> **On using Existing Foundational Models.** Given the large-scale multi-modal training of recent vision-language models and their generalization capabilities, there is a need for evaluation benchmarks going beyond simple common corruptions [1] or adversarial perturbations [2] to better understand their behavior in the wild. The main motivation of our proposed work is to study how changes in the object background can affect the performance of various uni/multi-modal models. To this end, the proposed framework utilizes existing foundational models such as, SAM [3] and BLIP-2 [4] for visual and textual conditioning. Recent relevant works also utilize large-scale pretrained models but for global perturbations, while ours specifically studies the background changes impact on models’ performance. For instance, LANCE [5] (NeurIPS’23) employs the use of pretrained Large Language Models alongside an already proposed image editing technique [6] to induce global modifications in the real images. Such works underscore a shared trend in the field: the creative re-purposing of existing foundational models to forge new ways to evaluate the robustness of uni/multi-modal models. Different from LANCE, our contribution lies in how we have utilized the complementary strengths of different foundational models for our proposed object-to-background benchmarking. Our work preserves the semantics, shape, angle, and color of the original object while changing its composition to the background in the real images.  By combining the capabilities of current image-to-text, image-to-segment and text-to-image-models, we have created a framework that is significantly more adept at handling complex background changes. Furthermore, we use our framework to generate adversarial background changes by optimizing the visual and textual latents of the diffusion model. Our rigorous set of experiments on vision tasks such as, image classification, object detection and segmentation are designed to validate the effectiveness of our framework across uni/multi-modal models.
>
> **Generating Diverse Backgrounds.** Kindly note that in Figure 3 (main paper), we generate diverse, realistic background changes by a variety of prompts using ChatGPT rather than random seeds. We provide these ChatGPT prompts in Table 7 (revised Appendix A.3). This demonstrates the effectiveness of our text prompt-based diversity in image generation. Additionally, we provide further qualitative results of this capability in the appendix, specifically in Figures 8 and 9 (see revised Appendix A.3), where we display a range of results achieved using these varied text prompts.
>
> We show the effect of random seeds in Figure 7 in Appendix A.3, where we use BLIP-2 captions for the original image as a text prompt. We fixed this text prompt and generated images with different random seeds. We observe from rows 2, 3 and 5 of Figure 7, that altering the seed while keeping the same caption does not yield significant background diversity across the generated samples. Furthermore, using random seeds only does not provide the fine-grain control needed to benchmark vision-based models, which is our main objective. These observations highlight the importance of our designed textual prompts to generate diverse object-to-background changes for benchmarking current vision-based models.
>
> **On Studying Robustness of Vision-Language Models.**  In this work, our focus has been on examining vision language models through various lenses, including classification, detection, segmentation, and image captioning. Specifically, we have conducted evaluations of vision-language models like CLIP and EVA-CLIP for zero-shot classification tasks (Table 2 in the main paper, Table 10 in the revised Appendix). For the case of segmentation, our attention has been on assessing the performance of the multi-modal segmentation model SAM (Table 8 in the revised Appendix A.6). Moreover, for the task of image captioning, we have evaluated the robustness of the image-to-text BLIP-2 model (Table 4 in the main paper). Along with these, we study different uni-modal CNNs and vision transformers trained naturally and adversarially. As suggested, we will modify the title to avoid any confusion such as "Backbench:  Are Vision Models Resilient to Object-to-Background Context?"

---

> ### Author Response · Authors · 2023-11-22
> **Response to Reviewer SPdM (2/3)**
>
> **Evaluating Recent Vision Models.** As recommended, we have conducted evaluations on recent models such as DeiT[7] and ConvNeXt[8] in the Table below (also added in Section A.13 in the revised Appendix). We observe a consistent trend of decrease in performance to background changes across these models as well. We observe that DeiT-based models which distill the knowledge from a strong CNN-based model during training show overall improved performance across our object-to-background changes compared to ViT-based models (please see Table 1 in main paper).
>
>
> |                      |        | DeiT   |        |         |            | CNN        |            |         |
> |:--------------------|:------:|:------:|:------:|:-------:|:----------:|:----------:|:----------:|:-------:|
> | Background (IN-Nat)  | DeiT-T | DeiT-S | DeiT-B | **Average** | ConvNeXt-T | ConvNeXt-B | ConvNeXt-L | **Average** |
> | Original             | 96.36  | 99.27  | 99.41  | 98.34   | 99.07      | 99.21      | 99.40      | 99.22   |
> | Class Label          | 94.18  | 96.85  | 97.74  | 96.25   | 97.60      | 97.51      | 97.51      | 97.54   |
> | BLIP-2 Caption       | 89.33  | 94.29  | 95.07  | 92.89   | 94.64      | 94.82      | 95.47      | 94.97   |
> | Color                | 80.96  | 89.48  | 91.11  | 87.13   | 92.11      | 93.58      | 93.58      | 93.09   |
> | Texture              | 74.15  | 84.01  | 86.75  | 81.63   | 88.50      | 89.50      | 91.13      | 89.71   |
> | Background (IN-Adv)  |        |        |        |         |            |            |            |         |
> | Original             | 95.44  | 99.10  | 99.10  | 97.88   | 99.00      | 99.00      | 92.92      | 96.97   |
> | Adversarial          | 20.40  | 29.62  | 34.81  | 28.27   | 32.88      | 42.52      | 48.60      | 41.33   |
>
> *Comparison of recent CNN and transformer-based models (Top-1%) against our proposed object-background changes our IN-Nat and IN-Adv datasets. The table is also added in revised Appendix A.13 (Table 14).*
>
> Furthermore, we also provide results on DINOv2-based models [9]. Our preliminary experiments with the DINOv2 models, as presented in Table below (see A.14 in revised Appendix), show that DINOv2 models which utilize registers (learnable tokens) [10] during training for learning more interpretable attention maps, provide more robustness to background changes, with significant improvement seen in the adversarial background changes.
>
>
> |                      |       | DINOv2 |       |         |       | DINOv2 (with register) |       |         |
> |:---------------------|:-----:|:------:|:-----:|:-------:|:-----:|:---------:|:-----:|:-------:|
> | Background (IN-Nat)  | ViT-S | ViT-B  | ViT-L | **Average** | ViT-S | ViT-B                  | ViT-L | **Average** |
> | Original             | 96.78 | 97.18  | 98.58 | 97.51   | 97.71 | 98.05                  | 99.14 | 98.30   |
> | Class Label          | 94.62 | 96.02  | 97.18 | 95.94   | 95.55 | 96.44                  | 97.94 | 96.64   |
> | BLIP-2 Caption       | 89.33 | 94.29  | 95.07 | 92.89   | 94.64 | 94.82                  | 95.47 | 94.97   |
> | Color                | 83.85 | 89.68  | 93.31 | 88.94   | 85.88 | 91.15                  | 94.64 | 90.55   |
> | Texture              | 83.63 | 89.08  | 92.44 | 88.38   | 84.98 | 91.03                  | 93.97 | 89.99   |
> | Background (IN-Adv)  |       |        |       |         |       |                        |       |         |
> | Original             | 95.12 | 96.50  | 98.10 | 96.57   | 97.91 | 97.80                  | 99.00 | 98.23   |
> | Adversarial          | 54.31 | 71.62  | 80.87 | 68.93   | 58.30 | 76.21                  | 84.50 | 73.00   |
>
>
> *Evaluation of Dinov2-based models on IN-Nat and IN-Adv dataset.  DINOv2 model with registers shows higher robustness to background changes, particularly in the presence of color, texture, and adversarial backgrounds. The table is also added in revised Appendix A.14 (Table 15).*

---

> > ### Author Response · Authors · 2023-11-22
> > **Response to Reviewer SPdM (3/3)**
> >
> > **Results and Conclusions.** Our results section provides a comprehensive analysis revealing that vision-based models are vulnerable to background variations. We observe:
> >
> > 1. Vison-based models are vulnerable to diverse background changes, such as texture and color, with the **most prone to adversarial background changes** (Table 1 in the paper, Tables 10,11 and 14 in the Appendix).
> > 2. **Increasing the capacity of the model** for both CNN and transformer-based models helps in improving the robustness against varying background contexts. This indicates, that distilling from a more robust model can help in improving the robustness of small models (Table 1 in the main paper, Tables 10, 11, 14 and 15 in the revised Appendix).  Evidence of this is seen in the performance of DeiT-T, which, by distilling knowledge from a strong CNN-based model, shows improved robustness as compared to ViT-T (see Section A.13 in Appendix).
> > 3. Our study indicates that **adversarially trained models have limited robustness.** While they perform well in scenarios with adversarial background changes, their effectiveness is limited for other types of object-to-background compositions, as seen in Figure 4 of our main paper. This highlights a significant gap in current model training approaches when it comes to dealing with diverse background changes.
> > 4. Object detection and segmentation models, which incorporate object-to-background context, display reasonably better robustness to background changes than classification models, as evidenced by quantitative and qualitative results (Table 5 and Figure 5 in the paper and Section A.5 in the Appendix).
> > 5. Also, recent training approaches for vision transformer-based classification models that learn more interpretable attention maps [10] show improvement in robustness to background changes (Section A.14 in revised Appendix). This potentially indicates the importance of inducing object-to-background context during training as revealed by our work.
> > 6. Furthermore, as shown in Table 2 (paper), models trained on large-scale datasets with more scalable and stable training show better robustness against background variations.
> >
> > In conclusion, our research highlights the need to evolve vision-based models that are capable of accurately discerning pertinent features in diverse settings, shifting away from reliance on background elements. This approach not only aligns with recent innovations in the field but also paves the way for enhancing the adaptability and robustness of vision-based models.
> >
> > **Changes in Textual prompts.** Table 6 (Appendix) refers to the textual prompts used to generate background changes related to color, texture, and class name specifically. We also show the huge space of background changes that can be explored by showing qualitative samples in Figures 3 (main paper), 8, and 9 (Appendix), where we generate diverse realistic background changes using prompts varied generated using ChatGPT (see Table 7 in revised Appendix A.3).
> >
> >
> > **Minor Changes**
> > We thank the reviewer and have incorporated references to SAM and BLIP-2 in Section 3.2 as recommended and provided a reference in Section 3.2 for a clearer understanding of the changes made to the textual prompt. In our approach, for adversarial background changes, we optimize both visual and textual latents of the diffusion model while for non-adversarial background changes, we employ the default inpainting diffusion pipeline.
> >
> > Further, to improve clarity, we have revised Figure 2 in Section 3.1 of the main paper. Detailed insights and results mentioned above are now included in the appendix. Moreover, in the final revision, we plan to refine the results Section 4.1, aiming for greater clarity and ease of understanding.
> >
> > **References**
> >
> > [1] Hendrycs et al. "Benchmarking Neural Network Robustness to Common Corruptions and Perturbations", ICLR 2019.
> >
> > [2] Madry et al. "Towards deep learning models resistant to adversarial attacks", ICLR 2018.
> >
> > [3] Kirillov et al. "Segment anything.", ICCV, 2023.
> >
> > [4] Li et al. "Blip-2: Bootstrapping language-image pre-training with frozen image encoders and large language models." arXiv preprint arXiv:2301.12597 (2023).
> >
> > [5] Prabhu et.al "LANCE: Stress-testing Visual Models by Generating Language-guided Counterfactual Images", NeurIPS, 2023.
> >
> > [6] Hertz et.al "Prompt-to-prompt image editing with cross attention control", ICLR, 2022.
> >
> > [7] Touvron et al. "Training data-efficient image transformers & distillation through attention.", ICML, 2021
> >
> > [8] Liu et al. "A convnet for the 2020s.", CVPR, 2022
> >
> > [9] Oquab et al. "Dinov2: Learning robust visual features without supervision." arXiv preprint arXiv:2304.07193 (2023).
> >
> > [10] Darcet et al. "Vision Transformers Need Registers." arXiv preprint arXiv:2309.16588 (2023).

---

> ### Comment · Reviewer_SPdM · 2023-12-01
> **Official comment by Reviewer SPdM**
>
> Thank the authors for the feedback and new results.
>
> I still have concerns regarding the contributions of this work. The data generation approach are not new. Even with ChatGPT prompts, generating the diverse backgrounds with textual conditions has been explored before.
>
> Thanks again for the comments but I'm not fully convinced to raise my rating.

---

### Official Review · Reviewer_RPSV · 2023-10-31

**Soundness:** 4 excellent
**Presentation:** 4 excellent
**Contribution:** 3 good
**Rating:** 8
**Confidence:** 4

**Summary:**

This paper proposes an architecture to generate various images to evaluate the resilience of vision models. Specifically, the architecture proposed in this paper properly utilizes modern image-to-text, text-to-image, and image-to-segment models to generate various versions of an image with different backgrounds with same key object.

The experimental results included in this paper shows robustness  of various vision models against background changes, which show how a vision model understands images similar to humans.

**Strengths:**

The novelty of this paper comes from its efficiency and effectiveness. Since evaluation sets gathered from wild contains hidden correlation between the object and background inside an image, models trained with images-in-wild tend to rely on this correlation while humans don't.

As shown in experimental results, evaluating each vision model on various settings effectively tells the robustness of each model. This is where another strength of this paper comes from. While previous works focus on specific settings, the coverage of proposed method is broad.

**Weaknesses:**

While this paper shows broad coverage over robustness on object-background, I would say this cannot lead to the conclusion that a vision model is robust like humans. Further suggestion on possible future work will help readers understand the possibility of future development.

**Questions:**

As mentioned in weakness section, any suggestion on research direction will further promote future works in this area.
I also have a small concern on novelty of this work since the architecture proposed in this work can be seen as a mixture of existing methods. More justification on novelty of proposed architecture will strengthen the paper.

---

> ### Author Response · Authors · 2023-11-22
> **Response to Reviewer RSPV (1/3)**
>
> We thank the reviewer for the insightful comments. Please find our responses to specific queries below.
>
> **Future Directions.** Our research work is one of the preliminary efforts in utilizing diffusion models to study the object-to-background context in vision-based systems. Based on our observations and analysis, the following are the interesting future directions.
>
> 1. Since large capacity models in general show better robustness to object-to-background compositions, coming up with new approaches to effectively distill knowledge from these large models could improve how small models cope with background changes. This can improve resilience in small models that can be deployed in edge devices.
> 2. Another direction is to set up object-to-background priors during adversarial training to expand robustness beyond just adversarial changes. To some extent, successful examples are recent works [1, 2], where models are trained to discern the salient features in the image foreground. This leads to better robustness. Kindly see our detailed robustness analysis on DINO-v2 with registers [1] in our response to reviewer-**SPdM "Evaluating Recent Vision Models".** This shows that similar techniques can improve adversarial training as well.
> 3. Our work can be extended to videos where preserving the semantics of the objects across the frames while introducing changes to the background temporally will help understand the robustness of video models.
> 4. Additionally, the capabilities of diffusion models can be explored to craft complex changes in the object of interest while preserving the semantic integrity. For instance, in [3], diffusion models are employed to generate multiple viewpoints of the same object. Additionally, in [4], non-rigid motions of objects are created while preserving their semantics. By incorporating these with our approach, we can study how vision models maintain semantic consistency in dynamic scenarios.
>
> **On the Novelty of Proposed Architecture.** Given the large-scale multi-modal training of recent vision-language models and their generalization capabilities, there is a need for evaluation benchmarks going beyond simple common corruptions [5] or adversarial perturbations [6] to better understand their behavior in the wild. The main motivation of our proposed work is to study how changes in the object background can affect the performance of various uni/multi-modal models. To this end, the proposed framework utilizes existing foundational models such as, SAM [7] and BLIP-2 [8] for visual and textual conditioning. Recent relevant works also utilize large-scale pretrained models but for global perturbations, while ours specifically studies the background changes impact on models’ performance. For instance, LANCE [9] (NeurIPS’23) employs the use of pretrained Large Language Models alongside an already proposed image editing technique [10] to induce global modifications in the real images. Such works underscore a shared trend in the field: the creative re-purposing of existing foundational models to forge new ways to evaluate the robustness of uni/multi-modal models. Different from LANCE, our contribution lies in how we have utilized the complementary strengths of different foundational models for our proposed object-to-background benchmarking. Our work preserves the semantics, shape, angle, and color of the original object while changing its composition to the background in the real images.  By combining the capabilities of current image-to-text, image-to-segment and text-to-image-models, we have created a framework that is significantly more adept at handling complex background changes. Furthermore, we use our framework to generate adversarial background changes by optimizing the visual and textual latents of the diffusion model. Our rigorous set of experiments on vision tasks such as, image classification, object detection and segmentation are designed to validate the effectiveness of our framework across uni/multi-modal models.

---

> ### Author Response · Authors · 2023-11-22
> **Response to Reviewer RSPV (2/3)**
>
> **Comparison with recent State-of-the-art Method.**  We compare with the recently proposed LANCE [9] (NeurIPS’23), which is relevant to our approach for benchmarking robustness. Their work leverages the capabilities of large language models to create textual prompts, facilitating diverse image alterations using the prompt-to-prompt method [10] and null-text inversion [11] for real image editing. We observe the following noticeable difference with LANCE as compared to our proposed approach:
>
> 1. LANCE relies on prompt-to-prompt method [10] and null-text inversion [11] for real image editing which allows modification of only specific words in the textual prompt. This limitation can restrict the range of possible image transformations. In contrast, our method uses visual conditioning to better preserve the object semantics, allowing us the flexibility to use textual guidance to introduce diverse background changes.
> 2. LANCE makes global changes to a given input image which can alter object semantics, e.g., shape distortion (See Table 13 and Figures 25 and 26 in the revised Appendix A.9).
> 3. Furthermore, LANCE needs adjustments in attention-based manipulation hyperparameters, and the selection of optimal images based on CLIP scores for each sample which is not scalable to larger datasets. In contrast, our approach allows for the evaluation of vision-based models across a much broader range of vision tasks. On the task of classification, we provide an evaluation of 5.5k images across 582 classes of ImageNet (See Figure 32 in the revised Appendix A.12), as opposed to the 15 classes evaluated in LANCE.
>
> We use open-sourced code from LANCE to compare it against our approach both quantitatively and qualitatively. We use a subset of 1000 images, named IN-ADV, for comparison.
> **a)** Quantitative comparison: We observe that our natural object-to-background changes including color and texture perform favorably against LANCE, while our adversarial object-to-background changes perform significantly better as shown in the table below.
>
>
> | Method    | Object-to-background Change | ViT-T | ViT-S | Swin-T | Swin-S | Res-50 | Res-152 | DenseNet-161 | Average |
> |:----------|:----------------------------:|:-----:|:-----:|:------:|:------:|:------:|:-------:|:------------:|:-------:|
> | Original  | --                          | 95.5  | 97.5  | 97.9   | 98.3   | 98.5   | 99.1    | 97.2         | 97.71   |
> | LANCE [9] | --                          | 80.0  | 83.8  | 87.6   | 87.7   | 86.1   | 87.4    | 85.1         | 85.38   |
> | Ours      | Color                       | 67.1  | 83.8  | 85.8   | 86.1   | 88.2   | 91.7    | 80.9         | **83.37**   |
> | Ours      | Texture                     | 64.7  | 80.4  | 84.1   | 85.8   | 85.2   | 90.1    | 80.3         | **81.55**   |
> | Ours      | Adversarial                 | 18.4  | 32.1  | 25.0   | 31.7   | 2.0    | 28.0    | 14.4         | **21.65**   |
>
>
> *Quantitative comparison (Top-1%) of our approach with LANCE. Our approach performs favorably well for benchmarking the robustness of ViTs and CNNs. The table is also added in the revised Appendix A.9 (Table 12).*

---

> ### Author Response · Authors · 2023-11-22
> **Response to Reviewer RSPV (3/3)**
>
> **b)** Qualitative comparison: Since LANCE relies on global-level image editing, thus tends to alter the object semantics and distort the original object shape in contrast to our approach which naturally preserves the original object and alters the object-to-background composition only. This can be observed in qualitative examples provided in Figures 25 and 26 in revised Appendix section A.9. We further validate this effect by masking the original images and LANCE-generated counterfactual images. We observe an accuracy drop from 97.71% to 84.35% across different models. Note that this is the first step of our approach, where the background is just masked but not optimized for any background changes. However, when the background is masked in LANCE-generated counterfactual images, overall accuracy drops from **97.71%** to **71.57%**. This shows that the LANCE framework has changed the original object during optimization which is also reflected in Figures 25 and 26 in revised Appendix Section A.9. Therefore, our proposed approach allows us to study the correlation of object to background changes without distorting the original object.
>
> | Method    | Object-to-background Change | ViT-T | ViT-S | Swin-T | Swin-S | Res-50 | Res-152 | DenseNet-161 | Average |
> |:----------|:----------------------------:|:-----:|:-----:|:------:|:------:|:------:|:-------:|:------------:|:-------:|
> | Original  | --                          |  95.5 |  97.5 |   97.9 |   98.3 |   98.5 |    99.1 |         97.2 |   97.71 |
> | Original  | Masked Background           |  70.5 |  86.1 |   84.2 |   87.6 |   87.2 |    91.2 |         83.7 |   84.35 |
> | LANCE [9] | Masked Background           |  59.5 |  72.5 |   72.3 |   75.3 |   71.9 |    77.5 |         72.0 |   71.57 |
>
> *Quantitative evaluation (Top-1%) of LANCE on IN-Adv shows its tendency to alter the original object in contrast to our approach which preserves the original object during optimization. The table is also added in revised Appendix A.9 (Table 13).*
>
>
>
>
> **References**
>
> [1] Darcet et al. "Vision Transformers Need Registers." arXiv preprint arXiv:2309.16588 (2023).
>
> [2] Sitawarin et al. "Part-based models improve adversarial robustness.", ICLR, 2023.
>
> [3] Yuan et al. "CustomNet: Zero-shot Object Customization with Variable-Viewpoints in Text-to-Image Diffusion Models." arXiv preprint arXiv:2310.19784 (2023).
>
> [4] Kawar et al. "Imagic: Text-based real image editing with diffusion models.", CVPR, 2023.
>
> [5] Hendrycs et al. "Benchmarking Neural Network Robustness to Common Corruptions and Perturbations", ICLR 2019.
>
> [6] Madry et al. "Towards deep learning models resistant to adversarial attacks", ICLR 2018.
>
> [7] Kirillov et al. "Segment anything.", ICCV, 2023.
>
> [8] Li et al. "Blip-2: Bootstrapping language-image pre-training with frozen image encoders and large language models." arXiv preprint arXiv:2301.12597 (2023).
>
> [9] Prabhu et.al "LANCE: Stress-testing Visual Models by Generating Language-guided Counterfactual Images.", NeurIPS, 2023.
>
> [10] Hertz et.al "Prompt-to-prompt image editing with cross attention control", ICLR, 2022.
>
> [11] Mokady et al. "Null-text inversion for editing real images using guided diffusion models", CVPR, 2023.

---

### Official Review · Reviewer_B18W · 2023-11-01

**Soundness:** 2 fair
**Presentation:** 2 fair
**Contribution:** 2 fair
**Rating:** 6
**Confidence:** 3

**Summary:**

This work reports results on examining the resiliency to background change of standard image classification models. The background changes are generated by modifying object images through masked diffusion model inference. An input image is first segmented with the prompt of its corresponding object class. An image captioning model also generates a textual description for it. Then the textual description is altered to another textual prompt that aims to change the scene/background of the image. The diffusion model, trained for inpainting tasks, takes the altered textual prompt, the input image, and the segmentation mask to generate a new image with altered background pixels. The image classification models are tested on these generated images for their classification accuracy.

Results show that tested image classification models, either Transformer or CNN-based, show accuracy drops on the images with altered backgrounds. Object detection and instance segmentation seem to still function properly on these images.

**Strengths:**

- The initiative to use the diffusion model, with its strong image generation capacity, as a tool to further study the background problem in image classification models could be interesting to the ICLR audience.

- The pipeline of image manipulation seems reasonable, judged from the textual description.

- The results can support the conclusions from the previous papers that image classification models are in general overfitted to the background of the object while object detection models suffer less from this problem.

**Weaknesses:**

- My primary concern about the study, while already collaborative, is that there seems to be a lack of further analysis. For example, we do not know whether the image alteration process will create additional object information that will confuse the classification. Image classification models are only allowed to make one prediction for each image. Thus, any new visual objects created during the image manipulation could easily confuse them. To rule out this hypothesis, a causal analysis of the classification errors might be necessary. There could be other hypotheses that need to be ruled out, too.

**Questions:**

Please see the weakness section for the concern I have. My questions would be:

1) what are the potential "outside factors" could mislead us in the experimental results?
2) the term resiliency to object-to-background seems solely measured by the accuracy drop of classification models on images altered with the proposed method. Is this metric well-calibrated? Will there be another metric that can quantify this effect? Does this metric correlate to the resiliency of models against other background change processes? I am eager to hear the authors' responses to these questions.

---

> ### Author Response · Authors · 2023-11-22
> **Response to Reviewer B18W (1/2)**
>
> We thank the reviewer for the insightful comments. Please find our responses to specific queries below.
>
> **Potential External Factors and Experimental Results.** Kindly note that when composing object-to-background change with texture, color, or adversarial patterns, the target models can perceive those as some other class if that pattern or composition is dominant in that class during the training of the models.   We discuss the potential external factors and how our proposed approach minimizes the effect of those external factors during benchmarking.
>
> - **Possibility of extra objects in the Background:**  Kindly note that **a)** we use a pretrained diffusion model that is conditioned on a pretrained CLIP text encoder, this means that the generated output follows the latent space of the CLIP text encoder which is aligned with CLIP visual encoder. Therefore, we can measure the faithfulness of the generated sample w.r.t the textual prompt used to generate it. We can measure this by encoding the generated output and its corresponding text prompt within CLIP latent space. For a given sample, CLIP or EVA-CLIP performs zero-shot evaluation by measuring the similarity between embedding of class templates (e.g. 1000 templates of ImageNet class) with a given image. Thus, if we simply add the template for a textual prompt used to generate the object-to-background changes, then we can measure its alignment with the background changes. For instance, instead of using a “a photo of a {fish}” template for zero-shot classification, we add the **relevant template** that is with background change, such as “a photo of a {fish} in the vivid colorful background”.  In other words, the relevant template represents the object and background change we introduced. We validate this observation on the EVA-CLIP ViT-E/14+, a highly robust model. Using the class templates such as “a photo of a {}”, the model achieves 95.84% accuracy on the original images (IN-Nat dataset), which decreases to **88.33%** when our color background changes are applied. However, when using the **relevant template**,  the performance improves to **92.95%**, significantly reducing the gap between the performance on the original and color background changes from **7.51%** to **2.89%**. These results show that accuracy loss from background changes isn't due to unwanted background objects of other classes. Furthermore, we manually assess  2.89% of misclassified samples (very few samples, see Figures 28 and 29 in the revised Appendix A.10). These can be considered the hardest examples in our dataset. We observe that even in such hard cases the model's confusion often stemmed from the complex background patterns instead of the addition of unwanted objects. We observe a similar trend in the case of adversarial patterns as well (see Figure 30 in the revised Appendix A.10). **b)** Another empirical evidence of how our generated output closely follows the given textual prompts can be observed with BLIP-2 Caption of the original image. In this case, object-background change has similar results as compared to original images across different vision models (Table 2 in the main paper).

---

> ### Author Response · Authors · 2023-11-22
> **Response to Reviewer B18W (2/2)**
>
> - **Extension of Objects:**  As already detailed in Section A.11 and illustrated in Figure 31 of the revised Appendix, we encountered challenges **when dealing with objects that occupy a small region in the image,** sometimes leading to certain unwanted extensions to objects. To mitigate this, we filtered our dataset to focus on images where the object covers a significant area. **Additionally, we slightly expand object masks computed using SAM to better define boundaries and prevent object shape distortion in the background.**
>
>
> - **Preserving Object Semantics:** We preserve object semantics by using strong visual guidance via SAM for precise object delineation. In contrast, recent methods like LANCE [1] make global changes and inadvertently affect the object's semantics. Please refer to the revised Appendix A.9 (Figures 25 and 26) for a qualitative comparison. We also kindly refer to our response to reviewer-ATH5 **"Comparison with recent State-of-the-art Method."**
>
> The design choices discussed above, such as high-quality data filtering, strong visual guidance, and class-agnostic textual guidance, contribute to the well-calibrated results of our study. This indicates that our results using the conventional metrics such as classification accuracy are well calibrated as well in the context of our high quality of generated data as mentioned above. We note that these choices ensure that the models are primarily challenged by diverse changes in the background, rather than being misled by the presence of unwanted objects. This careful approach underlines the reliability of our findings and highlights the specific factors influencing model performance.
>
> **Our benchmark datasets will be made publicly available, to enable further exploration and build upon our work.**
>
>
> **References**
>
> [1] Prabhu et.al "LANCE: Stress-testing Visual Models by Generating Language-guided Counterfactual Images.", NeurIPS, 2023.

---

### Official Review · Reviewer_ATH5 · 2023-11-08

**Soundness:** 2 fair
**Presentation:** 3 good
**Contribution:** 3 good
**Rating:** 5
**Confidence:** 3

**Summary:**

This paper proposes a new benchmark for evaluating the resilience of current vision and vision-language models to object-to-background context on real images. The proposed BackBench utilizes the capabilities of image-to-text and image-to-segmentation foundational models to preserve the semantics and appearance of the object while adding diverse background changes in
real images through textual guidance of the diffusion model.

**Strengths:**

This article is well-written and easy to understand.

**Weaknesses:**

1. This article lacks innovation, and the proposed framework is a combination of existing methods, such as SAM, BLIP-2, and DDIM.
2. This article does not compare with the current SOTA method of adversarial and counterfactual manipulations.
2. An additional SAM model is used, which may not be a fair comparison with the SOTA methods.

**Questions:**

See Weaknesses

---

> ### Author Response · Authors · 2023-11-22
> **Response to Reviewer ATH5 (1/2)**
>
> We thank the reviewer for the insightful comments. Please find our responses to specific queries below.
>
> **On Novelty.** The main motivation of our proposed work is to study how changes in the object background can affect the performance of various uni/multi-modal models. To this end, the proposed framework utilizes existing foundational models such as, SAM [1] and BLIP-2 [2] for visual and textual conditioning. Recent relevant works also utilize large-scale pretrained models but for global perturbations, while ours specifically studies the background changes impact on models’ performance. For instance, LANCE [3] (NeurIPS’23) employs the use of pretrained Large Language Models alongside an already proposed image editing technique [4] to induce global modifications in the real images. Such works underscore a shared trend in the field: the creative re-purposing of existing foundational models to forge new ways to evaluate the robustness of uni/multi-modal models.
>
> Different from LANCE, our contribution lies in how we have utilized the complementary strengths of different foundational models for our proposed object-to-background benchmarking. Our work preserves the semantics, shape, angle, and color of the original object while changing its composition to the background in the real images.  By combining the capabilities of current image-to-text, image-to-segment and text-to-image-models, we have created a framework that is significantly more adept at handling complex background changes. Furthermore, we use our framework to generate adversarial background changes by optimizing the visual and textual latents of the diffusion model. Our rigorous set of experiments on vision tasks such as, image classification, object detection and segmentation are designed to validate the effectiveness of our framework across uni/multi-modal models.
>
> **Comparison with Recent State-of-the-art Method.**  As recommended, we compare with the recently proposed LANCE [3] (NeurIPS’23), which is relevant to our approach for benchmarking robustness. Their work leverages the capabilities of large language models to create textual prompts, facilitating diverse image alterations using the prompt-to-prompt method [4] and null-text inversion [5] for real image editing. We observe the following noticeable difference with LANCE as compared to our proposed approach:
>
> 1. LANCE relies on prompt-to-prompt method [4]  and null-text inversion [5] for real image editing which allows modification of only specific words in the textual prompt. This limitation can restrict the range of possible image transformations. In contrast, our method uses visual conditioning to better preserve the object semantics, allowing us the flexibility to use textual guidance to introduce diverse background changes.
> 2. LANCE makes global changes to a given input image which can alter object semantics, e.g., shape distortion (See Table 13 and Figures 25 and 26 in the revised Appendix A.9).
> 3. Furthermore, LANCE needs adjustments in attention-based manipulation hyperparameters, and the selection of optimal images based on CLIP scores for each sample which is not scalable to larger datasets. In contrast, our approach allows for the evaluation of vision-based models across a much broader range of vision tasks. On the task of classification, we provide an evaluation of 5.5k images across 582 classes of ImageNet (See Figure 32 in revised Appendix), as opposed to the 15 classes evaluated in LANCE.

---

> ### Author Response · Authors · 2023-11-22
> **Response to Reviewer ATH5 (2/2)**
>
> We use open-sourced code from LANCE to compare it against our approach both quantitatively and qualitatively. We use a subset of 1000 images, named IN-ADV, for comparison. **a)** Quantitative comparison: We observe that our natural object-to-background changes including color and texture perform favorably against LANCE, while our adversarial object-to-background changes perform significantly better as shown in the table below.
>
> | Method     | Object-to-background Change | ViT-T | ViT-S | Swin-T | Swin-S | Res-50 | Res-152 | DenseNet-161 | Average |
> |:-----------|:----------------------------:|:-----:|:-----:|:------:|:------:|:------:|:-------:|:------------:|:-------:|
> | Original   | --                          | 95.5  | 97.5  | 97.9   | 98.3   | 98.5   | 99.1    | 97.2         | 97.71   |
> | LANCE [3]  | --                          | 80.0  | 83.8  | 87.6   | 87.7   | 86.1   | 87.4    | 85.1         | 85.38   |
> | Ours       | Color                       | 67.1  | 83.8  | 85.8   | 86.1   | 88.2   | 91.7    | 80.9         | **83.37**   |
> | Ours       | Texture                     | 64.7  | 80.4  | 84.1   | 85.8   | 85.2   | 90.1    | 80.3         | **81.55**   |
> | Ours       | Adversarial                 | 18.4  | 32.1  | 25.0   | 31.7   | 2.0    | 28.0    | 14.4         | **21.65**   |
>
> *Quantitative comparison (Top-1%) of our approach with LANCE. Our approach performs favorably well for benchmarking the robustness of ViTs and CNNs. The table is also added in the revised Appendix A.9 (Table 12).*
>
>
> **b)** Qualitative comparison: Since LANCE relies on global-level image editing, thus tends to alter the object semantics and distort the original object shape in contrast to our approach which naturally preserves the original object and alters the object-to-background composition only. This can be observed in qualitative examples provided in Figures 25 and 26 in revised Appendix section A.9. We further validate this effect by masking the original images and LANCE-generated counterfactual images. We observe an accuracy drop from 97.71% to 84.35% across different models. Note that this is the first step of our approach, where the background is just masked but not optimized for any background changes. However, when the background is masked in LANCE-generated counterfactual images, overall accuracy drops from 97.71% to 71.57%. This shows that the LANCE framework has changed the original object during optimization which is also reflected in Figures 25 and 26 in revised Appendix Section A.9. Therefore, our proposed approach allows us to study the correlation of object to background changes without distorting the original object.
>
> | Method    | Object-to-background Change | ViT-T | ViT-S | Swin-T | Swin-S | Res-50 | Res-152 | DenseNet-161 | Average |
> |:----------|:----------------------------:|------:|------:|-------:|-------:|-------:|--------:|:-------------:|:--------:|
> | Original  | --                          |  95.5 |  97.5 |   97.9 |   98.3 |   98.5 |    99.1 |         97.2 |   97.71 |
> | Original  | Masked Background           |  70.5 |  86.1 |   84.2 |   87.6 |   87.2 |    91.2 |         83.7 |   84.35 |
> | LANCE [3] | Masked Background           |  59.5 |  72.5 |   72.3 |   75.3 |   71.9 |    77.5 |         72.0 |   71.57 |
>
> *Quantitative evaluation (Top-1%) of LANCE on IN-Adv shows its tendency to alter the original object in contrast to our approach which preserves the original object during optimization. The table is also added in revised Appendix A.9 (Table 13).*
>
>
>
>
> **On using SAM.** Kindly note that we use SAM [1] as our image-to-segment model for visual conditioning to *preserve the object semantics*. This aligns with our objective to study the robustness of vision-based models to object-to-background compositional changes. The requirement to preserve the object semantics needs an instance segmentation model like SAM, which if not used would result in editing the actual object (a limitation other global methods like LANCE suffer from).
>
>
>
>
> **References**
>
> [1] Kirillov et al. "Segment anything.", ICCV, 2023.
>
> [2] Li et al. "Blip-2: Bootstrapping language-image pre-training with frozen image encoders and large language models." arXiv preprint arXiv:2301.12597 (2023).
>
> [3] Prabhu et.al "LANCE: Stress-testing Visual Models by Generating Language-guided Counterfactual Images.", NeurIPS, 2023.
>
> [4] Hertz et.al "Prompt-to-prompt image editing with cross attention control", ICLR, 2022.
>
> [5] Mokady et al. "Null-text inversion for editing real images using guided diffusion models", CVPR, 2023.

---

### Author Response · Authors · 2023-11-22
**Thank you for the valuable comments**

We thank all the reviewers (ATH5, B18W, RPSV, SPdM) for the positive feedback and appreciate the detailed comments to improve our work. All changes will be reflected in our final manuscript. **Reviewer-ATH5:** "Article is well-written and easy to understand." **Reviewer-B18W:** "The Problem in image classification models could be interesting to the ICLR audience. The pipeline of image manipulation seems reasonable." **Reviewer-RPSV:** "The novelty of this paper comes from its efficiency and effectiveness. Since evaluation sets gathered from wild contains hidden correlation between the object and background inside an image, models trained with images-in-wild tend to rely on this correlation while humans don't.  Evaluating each vision model on various settings effectively tells the robustness of each model. This is where another strength of this paper comes from." **Reviewer-SPdM:** "This work focuses on an important topic about the robustness of vision and vision-language models to object-to-background context. Such benchmarks allow us to analyze the limitations of existing models."
**Our codes and the proposed generated datasets will be publicly released for benchmarking and reproducibility.**

Given the large-scale multi-modal training of recent vision-language models and their generalization capabilities, there is a need for evaluation benchmarks going beyond simple common corruptions [1] or adversarial perturbations [2] to better understand their behavior in the wild. Similar to ours, the recent work of [3] also explores leveraging large language models and diffusion models to generate counterfactual examples that may change the semantics of the original object in **real samples.** In contrast to [3], our work preserves the semantics, shape, angle, and color of the original object in real samples while changing its composition w.r.t the background. We achieve this by exploiting the complementary strengths of different foundational models. In this manner, our approach is able to better understand the vulnerability of different uni/multi-modal models to the change in correlation between the object of interest and its background. Our analysis reveals the following insights:



- Naturally trained models are most vulnerable to adversarial background changes while relatively less vulnerable to natural (color, texture) background changes as compared to adversarially trained models and vice versa (Table1 and Figure 4 in the main paper). This can potentially help in better design of adversarial training algorithms in the future.
-  Our analysis shows that the larger the model size, better the robustness for both CNN and transformer-based models against varying background context. This indicates that distilling from the larger model is likely to increase the robustness of small models (Table 1 in the main paper, Tables 10, 11, 14 and 15 in the revised Appendix).
- Object detection and segmentation models, which incorporate object-to-background context, display reasonably better robustness to background changes than classification models, as evident in our quantitative and qualitative results (Table 5 and Figure 5 in the main paper and Section A.5 in the Appendix). Similarly, recent training approaches for vision transformer-based classification models that learn more interpretable attention maps [4] show improvement in robustness to background changes (Section A.14 in revised Appendix). This potentially indicates the importance of inducing object-to-background context during training as revealed by our work.
- Our analysis also indicates that EVA-CLIP with more scalable and stable training than CLIP exhibits better resilience across different object-to-background changes (Table 2 in the main paper).


**References**

[1] Hendrycs et al. “Benchmarking Neural Network Robustness to Common Corruptions and Perturbations”, ICLR 2019.

[2] Madry et al. "Towards deep learning models resistant to adversarial attacks", ICLR 2018.

[3] Prabhu et al. "LANCE: Stress-testing Visual Models by Generating Language-guided Counterfactual Images", NeurIPS, 2023.

[4] Darcet et al. "Vision Transformers Need Registers." arXiv preprint arXiv:2309.16588 (2023).

---

### Meta-Review · Area_Chair_6RJL · 2023-12-06

**Metareview:**

In terms of strengths, reviewers found this work to be interesting, effective and have broad coverage of settings.

Reviewers' initial reviews raised valid concerns about novelty/innovation and comparison to other methods, usage of more recent CNN/Transformer models and limited analyses. To their credit, the authors added comparisons to an alternative SOTA method (LANCE), more results using DeiT and ConvNeXt, and explained the differences/advantage of the proposed approach relative to LANCE. Nonetheless, reviewers were only partially convinced.

After giving full consideration of the reviews, rebuttals and subsequent discussion -- and considering the broader novelty/significance of the work, I do not recommend acceptance of this work in its current form, as explained below.

Starting from the more detailed issues, the comparisons to LANCE are convincing for the adversarial changes, but less so for the color and texture changes. More broadly, more comparisons to other methods or baselines would make a more convincing case. I do not take too much issue with usage of off-the-shelf methods like SAM (unlike some reviewers), and I take on board the authors' explanations of novelty relative to LANCE -- but nonetheless the novelty is a little bit limited in scope. I view the work as a "system paper" with some nifty tricks (in the positive sense), but when viewed in a broader context, the main point (i.e. that many current models are not resilient to certain changes, especially adversarial ones) is not a particularly new finding.

Overall, for a highly selective venue like ICLR, this work falls a little short of the standards for acceptance. It is useful work, and I encourage the authors to continue improving the work, but I suspect that it would be much better received for a benchmark-specific submission track, or as a workshop paper.

**Justification For Why Not Higher Score:**

Limited scope and novelty. Comparisons to alternative methods or baselines need to be more thorough.

**Justification For Why Not Lower Score:**

N/A

---

### Decision · Program_Chairs · 2024-01-16

Reject